# Normalization-equivariant Diffusion Models: Learning Posterior Samplers From Noisy And Partial Measurements

**Brett Levac**[1]  **Jon Tamir**[1]  **Marcelo Pereyra**[2]  **Julián Tachella**[3]

## Abstract

Diffusion models (DMs) are a powerful framework for image generation and restoration. However, existing DMs are primarily trained in a supervised manner by using a large corpus of clean images. This poses fundamental challenges in many real-world scenarios, where acquiring noise-free data is hard or infeasible. While some methods are capable of training DMs using noisy data, they are effective only when the amount of noise is very mild or when additional noise-free data is available. In addition, existing methods for training DMs from incomplete measurements require access to multiple complementary acquisition processes, a significant practical limitation. Here we introduce the first approach for learning DMs for image restoration using only noisy measurement data from a single operator. First, we show that DMs, and more broadly minimum mean squared error denoisers, exhibit a weak form of scale equivariance linking rescaling in signal amplitude to changes in noise intensity. We then leverage this theoretical insight to develop a denoising score-matching strategy that generalizes robustly to noise levels below the training data, thereby enabling the learning of DMs from noisy measurements. For problems involving measurements both noisy and incomplete, we integrate our method with equivariant imaging, a complementary self-supervised learning framework that exploits the inherent invariants of imaging problems. This allows training DMs for image restoration from single-operator noisy measurements. We validate the effectiveness of our approach through extensive experiments on image denoising, demosaicing, inpainting, and MRI reconstruction along

with comparisons with the state of the art.

## 1. Introduction

Nearly all image data used in computer vision tasks come from a physical imaging system (e.g., digital camera, PET/CT scanner, etc.). These physical imaging systems span many disparate fields, from astronomy (Vojtekova et al., 2020) to medicine (Heckel et al., 2024). In each application, the crucial similarity is that the measurements from the imaging system are not always directly useful for downstream tasks. This can be due to the measurement process (e.g., mosaicing, blurring, Fourier encoding, etc.) and/or measurement noise.

Recovering the desired image $\mathbf{x} \in \mathbb{R}^n$ from (noisy) measurements $\mathbf{y} \in \mathbb{R}^m$ requires solving an inverse problem. This inversion process is typically ill-posed and some regularization is needed to obtain a robust inversion. Previously, handcrafted priors such as sparsity (Donoho, 2006; Lustig et al., 2007) were popular in many inversion schemes. With the growth of deep learning methods, many end-to-end networks for image recovery have been proposed (Hammernik et al., 2018; Aggarwal et al., 2019; Liang et al., 2021). Recently, there has been interest in using deep generative models (Bora et al., 2017; Kawar et al., 2022; Jalal et al., 2021; Chung et al., 2024; Holden et al., 2022; González et al., 2022; Spagnoletti et al., 2025) to compute estimators or sample from the posterior distribution $p(\mathbf{x}|\mathbf{y})$. Both settings rely strongly on supervised learning, where a large corpus of clean data $\{\mathbf{x}_i\}_{i=1}^N \sim p(\mathbf{x})$ is available for training.

In practice, however, we often do not have direct access to samples from the distribution $p(\mathbf{x})$ but rather to measurement samples $\{\mathbf{y}_i\}_{i=1}^N \sim p(\mathbf{y})$ where $\mathbf{y}_i$ are generated from the measurement process of the imaging system of interest. Prior works have proposed methods for learning a variety of estimators using only measurement data. These works range from end-to-end methods (Yaman et al., 2020; Moran et al., 2019; Lehtinen et al., 2018; Tachella et al., 2025a; Monroy et al., 2025) to generative techniques (Bora et al., 2018; Daras et al., 2023; 2024b;a; Lu et al., 2025). A key similarity between many of these techniques, however, is an assumption of either access to some noise-free data

[1]Department of Electrical and Computer Engineering, University of Texas at Austin, TX, USA [2]School of Mathematical and Computer Sciences, Heriot-Watt University [3]ENS de Lyon, France. Correspondence to: Brett Levac <blevac@utexas.edu>.

*Proceedings of the 43$^{rd}$ International Conference on Machine Learning*, Seoul, South Korea. PMLR 306, 2026. Copyright 2026 by the author(s).

and/or measurements arising from multiple imaging operators. These assumptions may not be met in practice, where all data are noisy and observed via a single operator.

Herein, we propose a method for learning generative diffusion models capable of restoring corrupted images using only degraded measurements, obtained from multiple images processed through a common measurement operator. Our first key contribution demonstrates that diffusion models (DMs), and more broadly minimum mean squared error (MMSE) denoisers, exhibit a weak form of scale equivariance, wherein rescaling the signal amplitude induces a corresponding change in the noise level. Leveraging this theoretical insight, we introduce a denoising score-matching framework that enables the learning of denoisers at noise levels below those present in the measurements. Because denoisers implicitly decompose images into multiple scales (Milanfar & Delbracio, 2024), these denoisers are likely taking advantage of the inherent variability in signal scale present within and across the training set. Our experiments on real-world, public datasets help corroborate this claim: by imposing scale equivariance together with SURE, we are able to get stronger denoising performance across different ambient noise levels compared to using SURE alone. These denoisers can then be integrated into a DM sampler, yielding reconstructions with higher perceptual quality compared to existing self-supervised methods. Furthermore, we extend our approach beyond denoising by incorporating equivariance to geometric transformations (e.g., translations and rotations), allowing us to learn DMs in the challenging setting where data are observed via a single incomplete measurement operator and corrupted by measurement noise.

## 2. Related Work

**Diffusion Modeling**  The primary goal of generative modeling techniques is to learn a probability distribution from observed samples, i.e., $\{\mathbf{x}_i\}_{i=1}^N \sim p(\mathbf{x})$. This can be accomplished in a variety of ways (Kingma & Welling, 2022; Goodfellow et al., 2014; Ho et al., 2020). After learning this distribution, the model can be queried to generate new samples from $p(\mathbf{x})$ via a sampling procedure. The most popular methods currently are diffusion based approaches (Ho et al., 2020; Song et al., 2021; Karras et al., 2022). These techniques accomplish the task by training a deep neural network with parameters $\boldsymbol{\theta}$ on the supervised loss,

$$\min_{\boldsymbol{\theta}} \sum_{i=1}^N \mathcal{L}_{\text{SUP}}(\mathbf{x}, \boldsymbol{\theta})$$
$$\text{where} \quad \mathcal{L}_{\text{SUP}}(\mathbf{x}, \boldsymbol{\theta}) = \mathbb{E}_{\boldsymbol{\eta}, \sigma_t} \| \mathrm{D}_{\boldsymbol{\theta}}(\mathbf{x} + \sigma_t \boldsymbol{\eta}, \sigma_t) - \mathbf{x} \|^2, \tag{1}$$

with $\boldsymbol{\eta} \sim \mathcal{N}(\mathbf{0}, \mathbf{I})$ and $\sigma_t \sim p(\sigma)$, such that the learned network approximates the MMSE estimator $\mathrm{D}_{\boldsymbol{\theta}}(\mathbf{x} + \sigma_t \boldsymbol{\eta}, \sigma_t) \approx$

$\mathbb{E}\{\mathbf{x}|\mathbf{x} + \sigma_t \boldsymbol{\eta}\}$ at each noise level $\sigma_t > 0$. The probability distribution can be sampled by solving the following stochastic differential equation (SDE) in reverse time,

$$d\mathbf{x} = -2\dot{\sigma}_t \frac{\mathrm{D}_{\boldsymbol{\theta}}(\mathbf{x}, \sigma_t) - \mathbf{x}}{\sigma_t} dt + \sqrt{2\dot{\sigma}_t \sigma_t} d\boldsymbol{\omega}_t, \tag{2}$$

where $\boldsymbol{\omega}_t$ is a Brownian noise process and $t \in (0, 1)$, using a variety of solvers (Karras et al., 2022).

**Self-Supervised Learning for Denoising**  When the data needed to train restoration and diffusion models come from a measurement device, there is no true notion of a clean, ground-truth training set (Belthangady & Royer, 2019; Lehtinen et al., 2018). The two primary challenges in learning models solely from corrupted measurements are noise and rank-deficient measurement operators. In the simplest denoising problem, the data are corrupted by additive Gaussian noise, i.e.,

$$\mathbf{y} = \mathbf{x} + \sigma_n \boldsymbol{\eta}, \quad \boldsymbol{\eta} \sim \mathcal{N}(\mathbf{0}, \mathbf{I}), \tag{3}$$

where $\sigma_n$ is called the *measurement* noise level. Self-supervised methods for learning denoisers from a dataset of noisy measurements alone, $\{\mathbf{y}_i\}_{i=1}^N$, have been developed and used with great success. For example, Stein's unbiased risk estimator (SURE) (Stein, 1981) provides an approach to learn an unbiased estimate of the MMSE estimator with access only to noise-corrupted data (Metzler et al., 2020; Soltanayev & Chun, 2021; Aali et al., 2023) by minimizing an equivalent objective which only relies on the measurements (where div is the divergence operator):

$$\mathcal{L}_{\text{SURE}}(\mathbf{y}, \boldsymbol{\theta}) = \| \mathbf{y} - \mathrm{D}_{\boldsymbol{\theta}}(\mathbf{y}, \sigma_n) \|^2 + 2\sigma_n^2 \operatorname{div} \mathrm{D}_{\boldsymbol{\theta}}(\mathbf{y}, \sigma_n). \tag{4}$$

A key limitation of SURE is that it can only be used to obtain an unbiased estimate of the supervised denoising loss in Equation (1) for noise levels at and above the measurement noise level (i.e., $\sigma_t \geq \sigma_n$). To learn diffusion processes with noisy data $\mathbf{y}$, however, we must be able to learn MMSE estimators below the measurement level $\sigma_t \leq \sigma_n$ as well. Recently, methods have been proposed to learn denoisers below the measurement noise level of the data (Daras et al., 2024b;a) by running the sampler during training and enforcing consistency, but they struggle when there are no clean data available to assist in training (Lu et al., 2025). In this paper, we show that *normalization-equivariance* can help bridge the gap to additionally learn below the measurement noise level without any clean data.

**Self-Supervised Learning for Linear Inverse Problems**  In the general linear inverse problem setting where data are corrupted by additive Gaussian noise, the measurements are given by

$$\mathbf{y} = \mathbf{A}\mathbf{x} + \sigma_n^2 \boldsymbol{\eta}, \quad \boldsymbol{\eta} \sim \mathcal{N}(\mathbf{0}, \mathbf{I}), \tag{5}$$

where $\mathbf{A} \in \mathbb{R}^{m \times n}$ is typically rank-deficient or highly ill-conditioned, and thus cannot be easily inverted. To overcome the rank-deficient operator, several end-to-end techniques have been proposed to train recovery methods from only corrupted measurement data (Yaman et al., 2020; Tachella et al., 2022), including generative models (Bora et al., 2018; Daras et al., 2023; Kawar et al., 2024; Kelkar et al., 2023; Park et al., 2025; Aali et al., 2025). Many of these techniques split the measurements and predict one subset of the measurements from a different subset of measurements from the same sample. These splitting techniques require access to measurement datasets $\{\mathbf{y}_i, \mathbf{A}_i\}_{i=1}^N$ with many different operators $\mathbf{A}_i \in \mathcal{A} \triangleq \{\mathbf{A}_1, \ldots, \mathbf{A}_G\}$. Intuitively, identifying a unique distribution $p(\mathbf{x})$ from measurements should be easier as $|\mathcal{A}|$ grows.

If, however, $|\mathcal{A}| = 1$ (i.e., there is only one measurement device) we must use additional information about the signal distribution $p(\mathbf{x})$ to assist in its recovery. The relatively mild assumption that the signal set of plausible images is invariant to a group of transformations $\{\mathbf{T}_g\}_{g=1}^G$, such as translations, flips and/or rotations, is often enough for learning from a single forward operator (Chen et al., 2022), as it allows us to create virtual operators in the following way:

$$\mathbf{y}_i = \mathbf{A}\mathbf{x} = \mathbf{A}\mathbf{T}_g^{-1}\mathbf{T}_g\mathbf{x} = \mathbf{A}_g\mathbf{x}', \qquad (6)$$

where $\mathbf{A}_g = \mathbf{A}\mathbf{T}_g$ and $\mathbf{x}' = \mathbf{T}_g\mathbf{x}$. In other words, we get $G$ virtual operators $\mathcal{A} = \{\mathbf{A}\mathbf{T}_1, \ldots, \mathbf{A}\mathbf{T}_G\}$ (Tachella et al., 2023). Invariance can be enforced in a self-supervised way using the equivariant imaging loss (Chen et al., 2022).

**Normalization Equivariant Denoisers** Previous lines of work have explored architecture changes for better noise level generalization in supervised denoisers (Mohan et al., 2020; Herbreteau et al., 2024). Recently, normalization-equivariant network architectures for image denoising were proposed in the supervised setting (Herbreteau et al., 2024). This work poses that certain conventional neural network components, such as ReLUs, should be removed and replaced with alternatives that retain normalization-equivariant properties over the network. This has been shown to lead to better generalization across noise levels in supervised training scenarios.

## 3. Normalization Invariant Denoising

In this section, we focus on the Gaussian denoising problem. We present a new self-supervised loss for learning denoisers below the measurement noise level, $\sigma_n$, using noisy data alone, $\{\mathbf{y}_i = \mathbf{x}_i + \sigma_n\boldsymbol{\eta}_i\}_{i=1}^N$. The resulting denoisers can subsequently be used to implement diffusion samplers.

As stated above, the MMSE denoiser can be learned at the measurement noise level without access to noise-free data by using SURE (4). However, SURE does not yield reliable

estimates for noise levels below that of the measurements; that is, it is only valid for $\sigma \geq \sigma_n$. To overcome this limitation, we assume that the denoiser satisfies the following normalization equivariance property:

$$\mathrm{D}_{\boldsymbol{\theta}}(\alpha\mathbf{y} + \mu\mathbf{1}, \sigma\alpha) = \alpha\,\mathrm{D}_{\boldsymbol{\theta}}(\mathbf{y}, \sigma) + \mu\mathbf{1}, \qquad (7)$$

for all $\alpha \in \mathbb{R}_+$, $\mu \in \mathbb{R}$ and $\mathbf{y}$ in the measurement distribution. This property relies on the assumption that any shifted and scaled version $\mathbf{x}' = \alpha\mathbf{x} + \mu\mathbf{1}$ of an image $\mathbf{x}$ should belong to the prior distribution, such that the scaled noisy input can be written as a noisy version of this scaled image as $\alpha\mathbf{y} + \mu\mathbf{1} = \mathbf{x}' + (\alpha\sigma)\boldsymbol{\eta}$ with noise level $\alpha\sigma$.

If Equation (7) is satisfied, then denoising at any noise level $\sigma'$ can be straightforwardly achieved by rescaling a single denoiser $\mathrm{D}_{\boldsymbol{\theta}}(\cdot, \sigma)$ trained at level $\sigma$, i.e., $\mathrm{D}_{\boldsymbol{\theta}}(\mathbf{y}, \sigma') = \frac{\sigma'}{\sigma}\mathrm{D}_{\boldsymbol{\theta}}(\frac{\sigma}{\sigma'}\mathbf{y}, \sigma)$. Equivariance can be enforced through architectural constraints as studied in (Herbreteau et al., 2024) with supervised denoising. Instead, we propose to embed the normalization equivariance property into the original SURE loss by using the modified loss

$$\begin{aligned}
\mathcal{L}_{\text{NE-SURE}}(\mathbf{y}, \boldsymbol{\theta}) = \\
\mathbb{E}_{\alpha,\mu} \big\{ &\|\alpha\mathbf{y} + \mu\mathbf{1} - \mathrm{D}_{\boldsymbol{\theta}}(\alpha\mathbf{y} + \mu\mathbf{1}, \alpha\sigma_n)\|^2 \\
&+ 2(\alpha\sigma_n)^2 \operatorname{div}\mathrm{D}_{\boldsymbol{\theta}}(\alpha\mathbf{y} + \mu\mathbf{1}, \alpha\sigma_n) \big\}, \quad (8)
\end{aligned}$$

where $\alpha \sim \mathcal{U}(0,1)$ and $\mu \sim \mathcal{U}(0,1)$, as we find that this leads to better performance for self-supervised learning, and where incorporating $\mu$ improves generalization. In practice, we evaluate the expectation by sampling a random pair $(\alpha, \mu)$, and use Monte Carlo SURE (Ramani et al., 2008) to approximate the divergence, requiring an additional network evaluation. Other approximations can be similarly employed under our equivariance assumption (Monroy et al., 2025).

**Posterior Sampling** After training a self-supervised, or supervised, denoiser we can sample from the posterior of noised measurement by initializing a reverse-time diffusion SDE processes at the measurement noise level $\sigma_n$ and running the SDE solver on Equation (2) to some $\sigma_{\min}$.

**Understanding Normalization Invariance** A valid question to ask is: Are there realistic priors whose associated MMSE denoisers verify this assumption? An initial answer to this question can be investigated by looking at the definition of a scale equivariant MMSE estimator:

$$\mathbb{E}(\mathbf{x}|\alpha\mathbf{y}, \alpha\sigma_n) = \frac{\int_{\mathbb{R}^n} p(\mathbf{x})p(\alpha\mathbf{y}|\mathbf{x}, \alpha\sigma_n)\mathbf{x}\,\mathrm{d}\mathbf{x}}{\int_{\mathbb{R}^n} p(\tilde{\mathbf{x}})p(\alpha\mathbf{y}|\tilde{\mathbf{x}}, \alpha\sigma_n)\,\mathrm{d}\tilde{\mathbf{x}}} \qquad (9)$$

$$= \frac{\int_{\mathbb{R}^n} p(\mathbf{x})p(\mathbf{y}|\frac{\mathbf{x}}{\alpha}, \sigma_n)\mathbf{x}\,\mathrm{d}\mathbf{x}}{\int_{\mathbb{R}^n} p(\tilde{\mathbf{x}})p(\mathbf{y}|\frac{\tilde{\mathbf{x}}}{\alpha}, \sigma_n)\,\mathrm{d}\tilde{\mathbf{x}}} \qquad (10)$$

$$= \frac{1}{\alpha}\frac{\int_{\mathbb{R}^n} p(\alpha\mathbf{x})p(\mathbf{y}|\mathbf{x}, \sigma_n)\mathbf{x}\,\mathrm{d}\mathbf{x}}{\int_{\mathbb{R}^n} p(\alpha\tilde{\mathbf{x}})p(\mathbf{y}|\tilde{\mathbf{x}}, \sigma_n)\,\mathrm{d}\tilde{\mathbf{x}}} \qquad (11)$$

where the first line uses that $p(\alpha\mathbf{y}|\mathbf{x}, \alpha\sigma_n) = p(\mathbf{y}|\frac{\mathbf{x}}{\alpha}, \sigma_n)$ for Gaussian noise, and the second line relies on a change of variables. Inspecting Equation (11), we see that choosing a positively homogeneous prior of order $k \in \mathbb{Z}$, i.e., if $p(\alpha\mathbf{x}) = \alpha^k p(\mathbf{x})$ for all $\mathbf{x} \in \mathbb{R}^n$ and all $\alpha > 0$, results in the normalization-equivariant condition in Equation (7) for $\mu = 0$ (this result can be extended to $\mu \neq 0$ by additionally assuming that $p(\mathbf{x})$ is invariant under additive shifts).

Unfortunately, no proper prior can be positively homogeneous, since we would get $\int_{\mathbb{R}^n} p(\mathbf{x})d\mathbf{x} = \infty$. Nonetheless, as shown in the theorem below, there are well-defined priors whose MMSE estimators are close to being normalization-equivariant. Again, for clarity, we focus on the case $\mu = 0$; the extension to $\mu \neq 0$ follows directly.

**Theorem 3.1.** *Let* $\mathrm{D}(\mathbf{y}, \sigma)$ *be the MMSE estimator to recover an unknown image* $\mathbf{x} \sim p(\boldsymbol{x})$ *from* $\mathbf{y} = \mathbf{x} + \sigma\boldsymbol{\eta}$ *with* $\boldsymbol{\eta} \sim \mathcal{N}(\mathbf{0}, \mathbf{I})$. *Assume that* $p(\mathbf{x})$ *admits a factorization* $p(\mathbf{x}) = p_1(\mathbf{x})p_2(\boldsymbol{x})$, *with* $p_1(\mathbf{x})$ *and* $p_2(\boldsymbol{x})$ *depending only on* $\|\mathbf{x}\|$ *and* $\mathbf{x}/\|\mathbf{x}\|$ *respectively, such that the normalized image* $\boldsymbol{x}/\|\mathbf{x}\|$ *is independent of* $\|\mathbf{x}\|$. *Also assume that* $U(\mathbf{x}) = -\log p(\mathbf{x})$ *is twice continuously differentiable. Then,* $\exists \sigma^\star > 0$ *such that* $\forall \sigma, \sigma' \in (0, \sigma^\star)$, $\mathrm{D}(\mathbf{y}, \sigma)$ *and* $\mathrm{D}(\mathbf{y}, \sigma')$ *are approximately equivalent by rescaling, i.e.,*

$$\| \mathrm{D}(\mathbf{y}, \sigma') - \tfrac{\sigma'}{\sigma} \mathrm{D}(\tfrac{\sigma}{\sigma'}\mathbf{y}, \sigma)\| \leq \epsilon,$$

*where the error* $\epsilon > 0$ *depends on the degree of non-homogeneity of* $p_1$, *as measured by the Fisher divergence w.r.t. to* $p(\mathbf{x}|\mathbf{y}, \sigma)$ *between* $p_1$ *and the closest positively homogeneous function.*

*Moreover, if* $p(\mathbf{x})$ *does not admit the above factorization, then approximate equivalence via rescaling holds with*

$$\| \mathrm{D}(\mathbf{y}, \sigma') - \tfrac{\sigma'}{\sigma} \mathrm{D}(\tfrac{\sigma}{\sigma'}\mathbf{y}, \sigma)\| \leq \epsilon'.$$

*for some* $\epsilon' > \epsilon$, *related to the distance between* $p(\mathbf{x})$ *and its closest distribution admitting the required factorization.*

*Proof.* Let $\mathcal{P}(\mathbb{R}^n)$ be the class of functions on $\mathbb{R}^n$ that are positively homogeneous and whose logarithm is twice continuously differentiable with Lipschitz continuous gradient. First, assume that $p(\mathbf{x})$ admits a factorization $p(\mathbf{x}) = p_1(\mathbf{x})p_2(\mathbf{x})$, with $p_1(\mathbf{x})$ and $p_2(\mathbf{x})$ depending only on $\|\mathbf{x}\|$ and $\mathbf{x}/\|\mathbf{x}\|$ respectively, and let $\tilde{p}_1$ be the function in $\mathcal{P}(\mathbb{R}^n)$ that is closest to $p_1$ in the sense of the Fisher divergence w.r.t. the posterior $p(\mathbf{x}|\mathbf{y}, \sigma) \propto p(\mathbf{x})p(\mathbf{y}|\mathbf{x}, \sigma)$, i.e.,

$$\tilde{p}_1 = \arg\min_{q \in \mathcal{P}(\mathbb{R}^n)} \int_{\mathbb{R}^n} \|\nabla \log q(\mathbf{x}) - \nabla \log p_1(\mathbf{x})\|^2 p(\mathbf{x}|\mathbf{y}, \sigma)d\mathbf{x}$$

Consider the approximation $\tilde{p}(\mathbf{x}) \propto \tilde{p}_1(\mathbf{x})p_2(\mathbf{x})$ of $p$, obtained by replacing the correct marginal $p_1$ by $\tilde{p}_1$, and denote by $\tilde{p}(\mathbf{x}|\mathbf{y}, \sigma) \propto \tilde{p}(\mathbf{x})p(\mathbf{y}|\mathbf{x}, \sigma)$ the associated posterior

distribution. We view $\tilde{p}$ as an operational prior that may be improper, but we assume that $\tilde{p}(\mathbf{x}|\mathbf{y}, \sigma)$ is well defined. Moreover, we denote by $\kappa_\sigma$ the Fisher divergence between the posteriors $p(\mathbf{x}|\mathbf{y}, \sigma)$ and $\tilde{p}(\mathbf{x}|\mathbf{y}, \sigma)$, given by

$$\kappa_\sigma = \int_{\mathbb{R}^d} \|\nabla \log p_1(\mathbf{x}) - \nabla \log \tilde{p}_1(\mathbf{x})\|^2 \pi(\mathbf{x}|\mathbf{y}, \sigma)d\mathbf{x},$$

where we have used the factorization property of $p$ and $\tilde{p}$ and Bayes' rule to simplify $\nabla \log p(\mathbf{x}|\mathbf{y}, \sigma) - \nabla \log \tilde{p}(\mathbf{x}|\mathbf{y}, \sigma)$.

Furthermore, because $\mathbf{x} \mapsto \log p(\mathbf{y}|\mathbf{x}, \sigma)$ is $1/\sigma^2$-strongly concave, and the Hessian of $\log \tilde{p}$ is bounded, there exists some $\sigma^\star$ such that for all $\sigma \leq \sigma^\star$ the approximation $\log \tilde{p}(\boldsymbol{x}|\boldsymbol{y}, \sigma)$ is strongly concave outside some compact set (i.e., for some constants $K > 0$ and $R \geq 0, \nabla^2 \log \tilde{p}(\mathbf{x}|\mathbf{y}) \succeq K\mathbf{I}$ for all $\|\mathbf{x}\| \geq R$). From (Huggins et al., 2018, Theorem 5.3), this implies that for any $\sigma \leq \sigma^\star$, the 2-Wasserstein distance between $p(\mathbf{x}|\mathbf{y}, \sigma)$ and $\tilde{p}(\mathbf{x}|\mathbf{y}, \sigma)$ is bounded as

$$\mathcal{W}_2 \left(p(\mathbf{x}|\mathbf{y}, \sigma), \tilde{p}(\mathbf{x}|\mathbf{y}, \sigma)\right) \leq \psi\kappa_\sigma,$$

where $\psi > 0$ depends on $\tilde{p}(\mathbf{x}|\mathbf{y}, \sigma)$, but is independent of $p(\mathbf{x}|\mathbf{y}, \sigma)$. For example, when $\tilde{p}(\mathbf{x}|\mathbf{y}, \sigma)$ is strongly log-concave, $\psi$ is the inverse of the log-concavity constant.

Following on from this, we denote by $\tilde{\mathrm{D}}(\mathbf{y}, \sigma)$ the MMSE denoiser associated with $\tilde{p}(\mathbf{x}|\mathbf{y}, \sigma)$ and use Equation (11) to show that $\tilde{\mathrm{D}}(\mathbf{y}, \sigma)$ verifies the desired rescaling property $\tilde{\mathrm{D}}(\mathbf{y}, \sigma') = \frac{\sigma'}{\sigma}\tilde{\mathrm{D}}(\frac{\sigma}{\sigma'}\mathbf{y}, \sigma)$. Lastly, because $\mathcal{W}_2$ bounds the difference in the expectation of random variables, we have

$$\|\mathrm{D}(\mathbf{y}, \sigma') - \tilde{\mathrm{D}}(\mathbf{y}, \sigma')\| \leq \psi\kappa_{\sigma'},$$
$$\|\tfrac{\sigma'}{\sigma}\mathrm{D}(\tfrac{\sigma}{\sigma'}\mathbf{y}, \sigma) - \tfrac{\sigma'}{\sigma}\tilde{\mathrm{D}}(\tfrac{\sigma}{\sigma'}\mathbf{y}, \sigma)\| \leq \tfrac{\sigma'}{\sigma}\psi\kappa_\sigma,$$

and therefore,

$$\|\mathrm{D}(\mathbf{y}, \sigma') - \tfrac{\sigma'}{\sigma}\mathrm{D}(\tfrac{\sigma}{\sigma'}\mathbf{y}, \sigma)\| \leq \epsilon,$$

with $\epsilon^2 = \psi^2\kappa_{\sigma'}^2 + (\frac{\sigma'}{\sigma})^2\psi^2\kappa_\sigma^2$, concluding the first part of the proof. The extension to cases where $p(\mathbf{x})$ cannot be factorized exactly is presented in the Appendix. This extension relies on introducing an intermediate approximation of $p(\mathbf{x})$ that does admit the required factorization, and then using this approximation in a triangle inequality argument. This further approximation leads the additional error $\epsilon' > \epsilon$. Please see the Appendix for details. □

To develop an intuition for Theorem 3.1, it is helpful to consider the following. In most imaging problems, the image magnitude $\|\mathbf{x}\|$ is primarily determined by ambient conditions such as illumination and instrumental sensitivity settings, and conveys negligible information about the normalized image $\mathbf{x}/\|\mathbf{x}\|$. Indeed, recovering $\mathbf{x}$ from the

total energy $\|\mathbf{x}\|^2$ alone is not meaningful and typically impossible. Therefore, the assumption that $p(\mathbf{x})$ admits the proposed factorization $p(\mathbf{x}) = p_1(\mathbf{x})p_2(\mathbf{x})$ is practically justified, at least as an operational approximation. In addition, it has been repeatedly empirically observed that the spectral properties of images exhibit power-law statistics related to self-similarity (see, e.g., (Ruderman, 1997) and Figure 14 in Appendix A). This motivates the consideration of power-law models of the form $p_1(\mathbf{x}) = q(\mathbf{x})L(\mathbf{x})$ where $q$ is positively homogeneous, e.g., $q(\mathbf{x}) = \|\mathbf{x}\|^{-\beta}$, and $L(\mathbf{x})$ is some slow-varying function that predominantly influences $p_1(\mathbf{x})$ near the origin. In this case, minimizing the considered Fisher divergence leads to $\tilde{p}_1 = q$, and the remaining error, $\kappa_{\sigma_n} = \int_{\mathbb{R}^n} L(\mathbf{x})p(\mathbf{x}|\mathbf{y},\sigma)d\mathbf{x}$, is small if $L(\mathbf{x})$ is small in regions with high posterior mass. In light of Theorem 3.1, because training by score-matching is equivalent to minimizing the Fisher divergence w.r.t. $p(\mathbf{x}|\mathbf{y},\sigma)$, we view performing score-matching subject to the considered normalization property as a mechanism for learning the approximate MMSE denoiser $\tilde{D}$ rather than D, and we expect this denoiser trained directly from noisy data to generalize robustly to lower noise levels provided $\sigma_n \leq \sigma^\star$ in Theorem 3.1. The experiments reported in Section 4 provide strong evidence in favor of this interpretation.

In addition, Theorem 3.1 highlights that one need not assume scale invariance of $p(\mathbf{x})$ to obtain an approximately scale invariant denoiser. This weaker property only requires the approximation to hold in the range of values of $\|\mathbf{x}\|$ where the posterior distribution has most of its probability mass, which is quite narrow in practice as $\mathbf{y}$ contains significant information about $\|\mathbf{x}\|$. Our experiments show that natural and scientific image distributions (e.g., AFHQ, FFHQ, NBU, MRI) exhibit this weak form of invariance inherently, providing empirical backing to this assumption.

## 4. Denoising Experiments

**Baselines** We evaluate the performance of the normalization-equivariance strategy below the measurement noise level by comparing to **(i)** a denoiser trained in supervised fashion at all noise levels **(ii)** a self-supervised denoiser trained using SURE only at the measurement noise level (SURE) to understand how seeing only the measurement noise level generalizes performance (Metzler et al., 2020), and **(iii)** a recently proposed method for self-supervised denoising below the measurement noise level using consistency (CD) (Daras et al., 2024a).

**Training Details** We compare the performance of all denoisers on the task of denoising FFHQ ($128 \times 128$ pixels) training data that has been corrupted with $\sigma_n = 0.075$. On the AFHQ ($128 \times 128$ pixels) dataset, we perform experiments for varying noise levels $\sigma_n \in \{0.05, 0.075, 0.10\}$.

Finally, we use a panchromatic satellite imaging dataset (NBU) (Meng et al., 2020) ($128\times128$ pixels) corrupted with $\sigma_n = 0.075$. To investigate the performance of our technique with varying dataset sizes, we create various training set sizes of $N \in \{500, 1000, 5000, 15000\}$ for $\sigma_n = 0.05$ using the AFHQ dataset. We emphasize that for all datasets, we simulated noise only once for each image in the training dataset to most accurately emulate a real-world setting. Our model architecture is the UNet denoiser from (Song et al., 2021) with 55M parameters.

**Sampler Details** For experiments on diffusion sampling, after training each model, we run Algorithm 1 in the Appendix with $\mathbf{A} = \mathbf{I}$ from the measurement noise level to $\sigma_{\min} = 0.01$ (for experiments with $\sigma_n \in \{0.05, 0.075\}$) or $\sigma_{\min} = 0.02$ (for experiments with $\sigma_n = 0.10$). We use $K = 25$ steps for all sampling methods, and the timestep schedule following (Karras et al., 2022):

$$\sigma_i = (\sigma_{\max}^{\frac{1}{\gamma}} + \frac{i}{N-1}(\sigma_{\min}^{\frac{1}{\gamma}} - \sigma_{\max}^{\frac{1}{\gamma}}))^\gamma, \qquad (12)$$

with $\gamma = 7$ and $\sigma_{\max} = \sigma_n$.

We report all metrics over validation sets of 1000, 1500, 1800 images for FFHQ, AFHQ, and NBU datasets, respectively. To assess distortion in the reconstructed images, we report PSNR and SSIM (Wang et al., 2004), while for perceptual quality, we report LPIPS and FID. We provide a closer look at the effect of inference timestep schedules in the Appendix (see Figure 19).

**Denoising Performance** Figure 1 shows how each denoiser performs at noise levels different than the measurement noise level available at training time. Both SURE and CD perform poorly at noise levels below the measurement noise level. In contrast, our approach incurs only a small performance reduction with respect to the supervised case at noise levels below the training data noise, providing strong empirical support for the validity of the normalization invariance assumption. Moreover, for highly noisy training data, the gap between the proposed method and supervised learning at lower noise increases, but it remains the best method across self-supervised techniques. Figure 2 (top) show examples of how all self-supervised denoisers perform well at the measurement noise level compared to the supervised approach. However, we observe in Figure 2 (bottom) that, at noise levels below the measurement noise level, only our technique maintains a good performance compared to the supervised strategy. Finally, we show our loss-based enforcement of normalization equivariance outperforms architecture enforcement (Herbreteau et al., 2024) in Figure 21 and Figure 22.

**Blind Denoising** Our training loss can be extended to blind settings where $\sigma_n$ is unknown by leveraging UN-

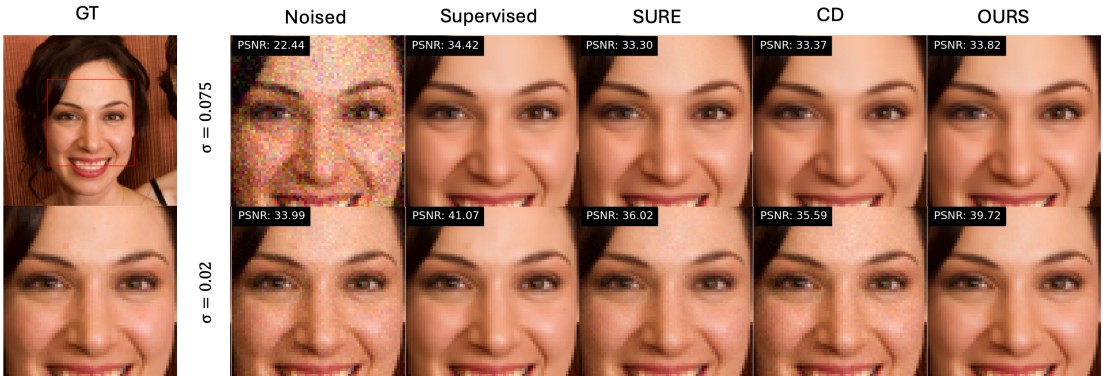

*Figure 1.* Performance of various one-step denoisers on FFHQ, NBU and AFHQ dataset.

*Figure 2.* Denoising example on FFHQ dataset at test noise levels $\sigma = 0.075$ (top) and $\sigma = 0.02$ (bottom). All models, except for supervised, were trained on only noisy data with $\sigma_n = 0.075$.

SURE (Tachella et al., 2025a) to construct an augmented loss that calibrates $\sigma_n$ automatically, see Equation (13). For details and results see the Appendix (Figure 15, Figure 16).

**Posterior Sampling Performance** We assess each denoiser's sampling ability compared to a supervised denoiser in Table 1. Here, MMSE (Self Sup.) is taken to be the one-step denoiser from our normalization-equivariant SURE denoiser, as it showed better PSNR metrics than SURE alone in the one-step denoising task above (i.e., better approximate self-supervised MMSE estimator). The sampler based on our self-supervised denoiser reports better perceptual quality metrics than all methods except for the supervised denoiser. As expected, all sampling approaches report worse PSNR and SSIM than their respective MMSE (one-step) solvers, which is in line with the known perception-distortion trade-off (Blau & Michaeli, 2018). In Figure 3 we show visual examples comparing each approach. Additionally, Figure 10 in Appendix A shows examples of each method at various training and inference noise levels on the AFHQ dataset along with each method's radial spectra for the corresponding reconstructions in Figure 11. All one-step solvers appear to reduce high frequency components resulting in a visually smoother image and CD retains noise in the sample leading to higher frequency components. Our method, however, produces spectra noticeably more closely aligned with the

ground truth images.

**Low Field MRI Denoising** To evaluate our training technique on real noise-corrupted data, we train a denoiser on an in-house dataset of low-field MRI wrist scans, where only raw noisy measurements are available. A denoised example is shown in Figure 24 where we see our denoiser retains good performance on real data. See Appendix for more details about the data collection process.

**Training Ablations** In Equation (8) we introduced $\alpha$ and $\mu$. To empirically show the importance of each term, we train various self-supervised denoisers on the AFHQ dataset with different sampling schemes for $\alpha$ and $\mu$ at training time. See Figure 17 in the Appendix for more details.

**Sample Complexity** Figure 4 shows the performance of our denoiser trained on the AFHQ dataset as a function of the number of (noisy) samples available for training. We plot the difference between the average MSE loss for each denoiser and the MMSE performance (approximated here by supervised learning on the largest dataset size). Previous work (Daras et al., 2024a) hypothesized that learning below the measurement noise level requires a potentially prohibitive amount of noisy training data. We find, on the contrary, that the mean squared error scales approximately

| Dataset | $\sigma_n$ | Solver | Sampler | Self Sup. | PSNR (↑) | SSIM (↑) | LPIPS (↓) | FID (↓) |
|---|---|---|---|---|---|---|---|---|
| FFHQ | 0.075 | MMSE (Self Sup.) | | ✓ | 33.68 | 0.937 | 0.024 | 29.82 |
| | | OURS | ✓ | ✓ | 32.76 | 0.921 | 0.015 | 22.57 |
| | | CD (Daras et al., 2024a) | ✓ | ✓ | 28.92 | 0.756 | 0.057 | 59.37 |
| | | MMSE (Sup.) (Karras et al., 2022) | | | **34.20** | **0.944** | 0.023 | 32.79 |
| | | EDM (Sup.) (Karras et al., 2022) | ✓ | | 33.06 | 0.920 | **0.012** | **19.91** |
| NBU | 0.075 | MMSE (Self Sup.) | | ✓ | 32.58 | 0.863 | 0.111 | 79.20 |
| | | OURS | ✓ | ✓ | 32.05 | 0.851 | 0.085 | 34.01 |
| | | CD (Daras et al., 2024a) | ✓ | ✓ | 32.02 | 0.836 | 0.076 | 53.75 |
| | | MMSE (Sup.) (Karras et al., 2022) | | | **33.08** | **0.877** | 0.110 | 103.18 |
| | | EDM (Sup.) (Karras et al., 2022) | ✓ | | 32.02 | 0.841 | **0.057** | **18.21** |
| AFHQ | 0.05 | MMSE (Self Sup.) | | ✓ | 34.52 | 0.949 | 0.020 | 10.09 |
| | | OURS | ✓ | ✓ | 33.89 | 0.942 | 0.011 | 5.08 |
| | | CD (Daras et al., 2024a) | ✓ | ✓ | 32.48 | 0.903 | 0.015 | 10.74 |
| | | MMSE (Sup.) (Karras et al., 2022) | | | **34.77** | **0.952** | 0.019 | 9.83 |
| | | EDM (Sup.) (Karras et al., 2022) | ✓ | | 34.01 | 0.942 | **0.010** | **3.62** |
| AFHQ | 0.075 | MMSE (Self Sup.) | | ✓ | 32.41 | 0.923 | 0.034 | 11.56 |
| | | OURS | ✓ | ✓ | 31.71 | 0.911 | 0.021 | 7.12 |
| | | CD (Daras et al., 2024a) | ✓ | ✓ | 29.44 | 0.818 | 0.034 | 17.08 |
| | | MMSE (Sup.) (Karras et al., 2022) | | | **32.72** | **0.928** | 0.034 | 12.55 |
| | | EDM (Sup.) (Karras et al., 2022) | ✓ | | 31.80 | 0.910 | **0.017** | **5.66** |
| AFHQ | 0.10 | MMSE (Self Sup.) | | ✓ | 30.96 | 0.898 | 0.053 | 13.80 |
| | | OURS | ✓ | ✓ | 30.38 | 0.888 | 0.034 | 8.88 |
| | | CD (Daras et al., 2024a) | ✓ | ✓ | 27.01 | 0.732 | 0.064 | 20.65 |
| | | MMSE (Sup.) (Karras et al., 2022) | | | **31.34** | **0.906** | 0.049 | 14.37 |
| | | EDM (Sup.) (Karras et al., 2022) | ✓ | | 30.26 | 0.879 | **0.024** | **7.82** |

*Table 1.* Posterior sampling metrics on FFHQ, NBU, and AFHQ $128 \times 128$ where the models are trained and complete inference on data with additive noise shown in the 2nd column.

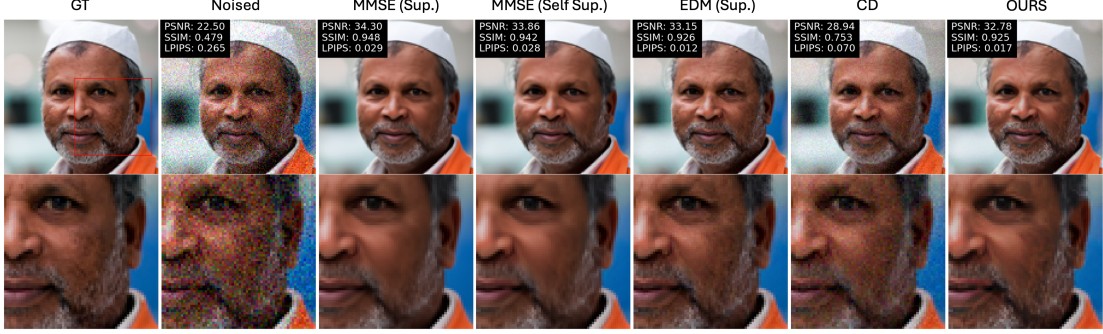

*Figure 3.* Example of various denoisers using diffusion sampling (except for MMSE (Self Sup.) and MMSE (Sup.) columns) on FFHQ dataset with training and test noise level $\sigma_n = 0.075$.

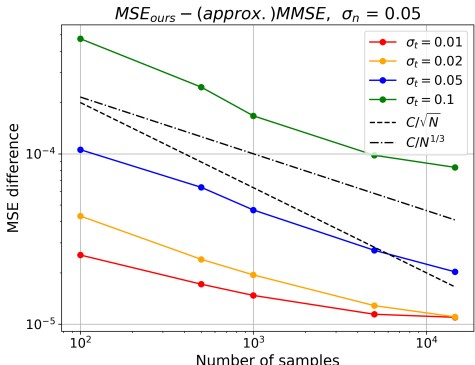

*Figure 4.* Denoiser performance on AFHQ with varying amounts of only noised training samples at $\sigma_n = 0.05$.

as $\sigma_t^2/N^{1/3}$ for noise levels above ($\sigma_t \geq 0.05$) and also below ($\sigma_t < 0.05$) the measurement noise, which is close to the complexity typically observed in supervised learning (Hardt & Recht, 2022, Ch. 6).

## 5. Extension to Linear Inverse Problems

We now extend our method to more general linear inverse problems. In particular, we present a new self-supervised loss for learning DMs from a dataset of noisy measurements, all observed by the same forward operator at the same measurement noise level, $\{\mathbf{y}_i = \mathbf{A}\mathbf{x}_i + \sigma_n \boldsymbol{\eta}_i\}_{i=1}^N$, where $\mathbf{A}$ is rank deficient and $\boldsymbol{\eta} \sim \mathcal{N}(\mathbf{0}, \mathbf{I})$.

Following the equivariant imaging (EI) approach, we assume that $p(\mathbf{x})$ is invariant to a group of transformations $\{\mathbf{T}_g \in \mathbb{R}^{n \times n}\}_{g=1}^G$ such as translations, flips and/or rota-

| Task | $\sigma_n$ | Solver | Sampler | Self Sup. | PSNR (↑) | SSIM (↑) | LPIPS (↓) | FID (↓) |
|---|---|---|---|---|---|---|---|---|
| | | EI (Chen et al., 2022) | | ✓ | **27.92** | **0.867** | 0.043 | 25.259 |
| Inpainting | 0.075 | OURS | ✓ | ✓ | 27.74 | 0.864 | **0.031** | **21.114** |
| | | Ambient Diff. (Daras et al., 2023) | ✓ | ✓ | 25.21 | 0.783 | 0.067 | 44.921 |
| | | EI (Chen et al., 2022) | | ✓ | **26.48** | **0.836** | 0.067 | 45.426 |
| Demosaic | 0.075 | OURS | ✓ | ✓ | 26.05 | 0.835 | **0.050** | **42.289** |
| | | Ambient Diff. (Daras et al., 2023) | ✓ | ✓ | 7.77 | 0.078 | 0.532 | 232.951 |

*Table 2.* Image inpainting (top) and Demosaicing (bottom) metrics on AFHQ $64 \times 64$.

tions, that is, for any image $\mathbf{x} \sim p(\mathbf{x})$, $\mathbf{T}_g\mathbf{x}$ follows the same distribution. We propose the following loss to approximately learn $\mathrm{D}_{\boldsymbol{\theta}}(\mathbf{y}, \sigma) \approx \mathbb{E}\{\mathbf{x}|\mathbf{Ax} + \sigma\boldsymbol{\eta}\}$ in a self-supervised fashion for all noise levels $\sigma \in (0, \sigma_n]$:

$$\mathcal{L}(\mathbf{y}, \boldsymbol{\theta}) = \mathbb{E}_{\alpha,\mu}\left\{\|\alpha\mathbf{y} + \mu\mathbf{1} - \mathrm{D}_{\boldsymbol{\theta}}(\alpha\mathbf{y} + \mu\mathbf{1}, \alpha\sigma_n)\|^2\right.$$
$$\left. + 2(\alpha\sigma_n)^2 \operatorname{div}\mathrm{D}_{\boldsymbol{\theta}}(\alpha\mathbf{y} + \mu\mathbf{1}, \alpha\sigma_n)\right\}$$
$$+ \mathbb{E}_{g,\sigma',\eta}\left\{\|\mathbf{T}_g\hat{\mathbf{x}}_{\boldsymbol{\theta}} - \mathrm{D}_{\boldsymbol{\theta}}(\mathbf{AT}_g\hat{\mathbf{x}}_{\boldsymbol{\theta}} + \sigma'\boldsymbol{\eta}, \sigma')\|^2\right\}$$

where $\hat{\mathbf{x}}_{\boldsymbol{\theta}} = \mathrm{D}_{\boldsymbol{\theta}}(\mathbf{y}, \sigma_n)$, $\sigma' \sim \mathcal{U}(0, \sigma_n)$, $\boldsymbol{\eta} \sim \mathcal{N}(\mathbf{0}, \mathbf{I})$. The first two terms are the same as in the denoising setting Equation (8) and allow generalizing to noise levels below the measurement level $\sigma_n$, and the last term is the EI loss (Chen et al., 2022) which allows learning in the nullspace of $\mathbf{A}$.

**Posterior Sampling** The learned network can be used to approximate the MMSE denoiser in measurement space as $\mathbb{E}\{\mathbf{Ax}|\mathbf{Ax} + \sigma\boldsymbol{\eta}\} = \mathbf{A}\mathbb{E}\{\mathbf{x}|\mathbf{y}\} \approx \mathbf{A}\mathrm{D}_{\boldsymbol{\theta}}(\mathbf{y}, \sigma)$. Thus, as in the denoising setting in Section 4, we can run the reverse SDE in Equation (2) in measurement space using the denoiser $\mathbf{A} \circ \mathrm{D}_{\boldsymbol{\theta}}$ from $\sigma_n$ to $\sigma_{\min}$, to sample an (almost) noiseless measurement $\mathbf{z} \sim p(\mathbf{Ax}|\mathbf{Ax} + \sigma_n\boldsymbol{\eta})$, and finally obtain an image space posterior sample as $\mathbf{x} = \mathrm{D}_{\boldsymbol{\theta}}(\mathbf{z}, \sigma_{\min})$. The resulting algorithm is summarized in Algorithm 1.

If the operator $\mathbf{A}$ is injective over the support of $p(\mathbf{x})$, that is, if reconstructions in the noiseless case are exact (see (Tachella et al., 2022) for technical details), then the final reconstruction step yields a sample of the correct posterior distribution $p(\mathbf{x}|\mathbf{y})$.

### 5.1. Experiments

**Baselines** We compare our approach to EI (Chen et al., 2022), a point estimator, with the same group of transformations. Additionally, we compare against Ambient Diffusion (Daras et al., 2023) trained without additive noise

**Training Details** We test the extension of our invariant sampler to non-trivial forward operators by solving inpainting, demosaicing, and MRI reconstruction tasks. For the inpainting task, we randomly simulated one inpainting mask with an undersampling factor of $m/n = 0.7$. After applying the same inpainting mask to all samples in the datasets, we added noise $\sigma_n = 0.075$ to the measurement data. Due to

the random structure of the inpainting mask, we use the translational invariance of natural images and choose $\mathbf{T}_g$ to represent all circular shifts of the image. For demosaicing, we prepare our training datasets by selecting the Bayer filter and applied the corresponding $\mathbf{A}$ to each image in the training dataset and added noise $\sigma_n = 0.075$. We assumed rotational invariance of the underlying signal set and selected the group of transformations $\mathbf{T}_g$ that represents all possible rotations. For both inpainting and demosaicing experiments, we used a training set size of 15000 from AFHQ ($64 \times 64$ pixels). For the MRI reconstruction task we provide more details and results in Appendix A.9. We utilize the DeepInverse (Tachella et al., 2025b) package for applying transformations in the equivariant imaging loss.

**Inference Details** For inference, we use Algorithm 1 with the same parameters as in Section 4, except we set $\mathbf{A}$ to the respective measurement operator.

**Sampler Performance** In Table 2, we show quantitative results comparing each self-supervised approach on inpainting and demosaicing. In both settings, EI retains superior distortion metrics (as expected for a point estimator analogous to an MMSE estimator (Blau & Michaeli, 2018)), but our approach beats both EI and Ambient Diffusion in perceptual quality metrics. Ambient Diffusion performs poorly because it lacks access to measurements from different measurement operators that cover the full ambient space. This means that at training time, there is a non-trivial nullspace that the model is never given self-supervised guidance to predict. We provide example restorations in the Appendix (Figures 12 and 13). We see similar results for MRI reconstruction which are shown in Figure 20.

## 6. Conclusion

We present a method for learning a denoiser from noisy data alone. In particular, the proposed method can learn below the noise level of the data, which allows constructing diffusion models for posterior sampling in image restoration. Our denoisers performs on par with supervised learning in noise levels below the measurement noise and are highly effective in diffusion sampling schemes. Moreover, our method can be extended to learn from noisy and incomplete measurements associated with a single degradation operator, leading to diffusion models that produce higher

perceptual quality images than existing self-supervised generative techniques. Future work will include extending this method to distilled diffusion models that allow few-step generation (Song et al., 2023), leveraging model architectures specific to image restoration (e.g., unrolled networks), conducting additional experiments on different modalities, and leveraging our approach for self-supervised fine-tuning of pre-trained foundation models to reduce distribution shift.

## Impact Statement

This paper presents work whose goal is to advance the field of Machine Learning. There are many potential societal consequences of our work, none which we feel must be specifically highlighted here.

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

# A. Appendix

## A.1. One-step Denoising Experiments

We have included additional figures for our one-step denoiser experiments. Figures 5 and 6 show the one step denoiser performance of the various methods at the measurement noise level and a lower test noise level for the AFHQ and NBU datasets respectively. We observe that our method performs better at denoising below the measurement noise level compared to other self-supervised denoising techniques. We have also included an ablation over training noise levels on the AFHQ dataset in Figure 7.

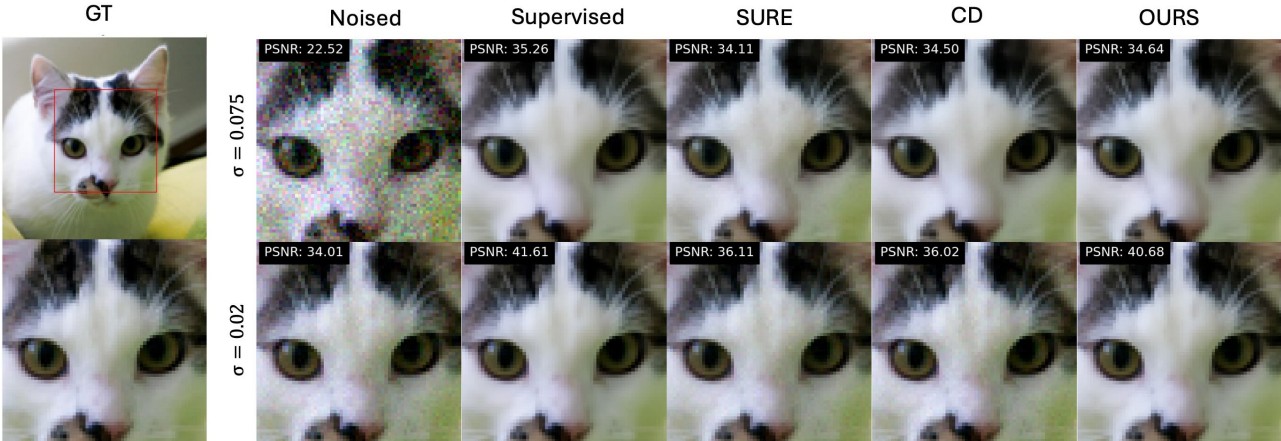

*Figure 5.* Example restorations of various denoisers on AFHQ dataset at test noise levels $\sigma_t = 0.075$ (top) and $\sigma_t = 0.02$ (bottom). All models, except for supervised were trained on only noisy data with $\sigma_n = 0.075$.

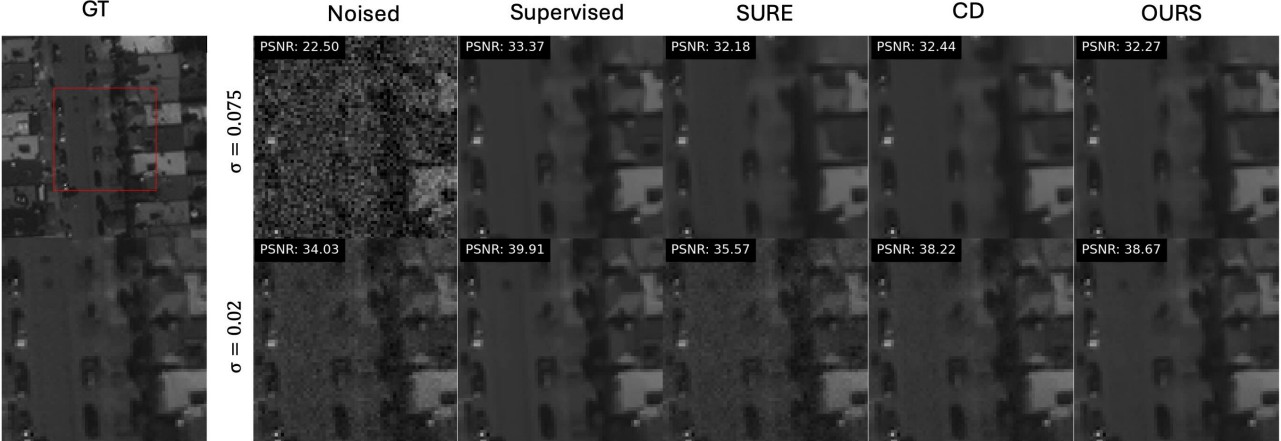

*Figure 6.* Example restorations of various denoisers on NBU dataset at test noise levels $\sigma_t = 0.075$ (top) and $\sigma_t = 0.02$ (bottom). All models, except for supervised were trained on only noisy data with $\sigma_n = 0.075$.

## A.2. Diffusion Sampling

Additional figures have been provided for diffusion sampling experiments on the AFHQ and NBU datasets. Figures 8 and 9 show example diffusion samples for supervised and self-supervised approaches discussed in the paper on AFHQ and NBU respectively. Figure 10 shows diffusion samples for different training + inference noise levels with accompanying radial spectrum plots in Figure 11. Here we see that while one-step supervised and self-supervised MMSE denoisers tend to reduce high frequency features, our method retains higher frequencies lending to our method providing better perceptual images.

## A.3. Extension to Linear Inverse Problems

We provide example reconstructions for the inpainting and demosaicing tasks in Figures 12 and 13 respectively.

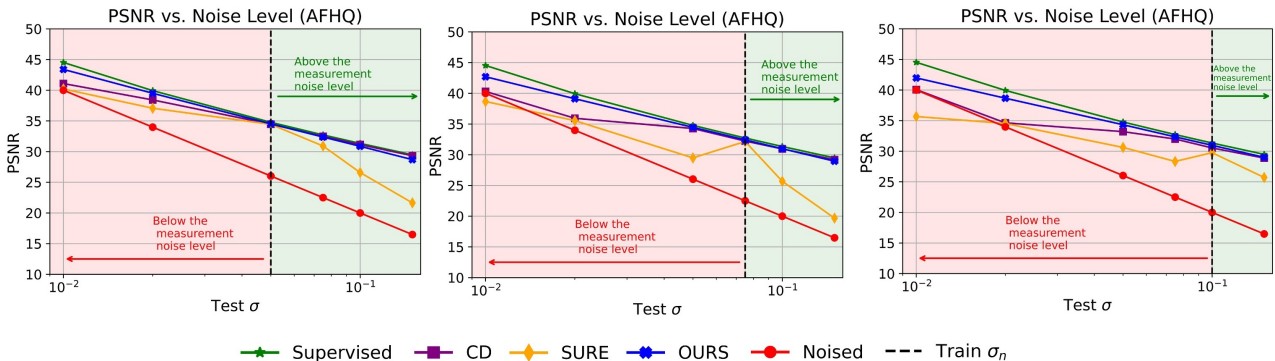

*Figure 7.* Performance of various one-step denoisers on AFHQ dataset with various training noise levels.

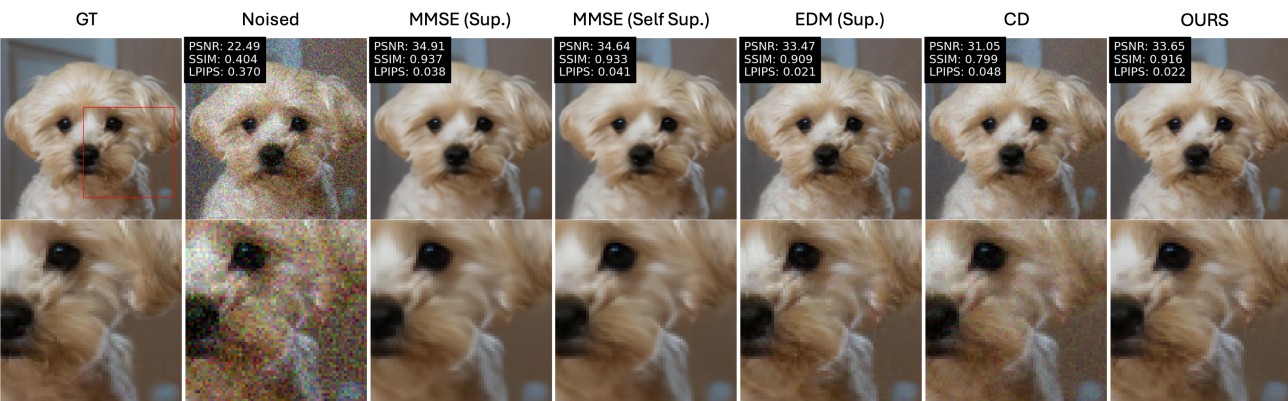

*Figure 8.* Example of various denoisers using diffusion sampling (except MMSE(Self Sup.) and MMSE(Sup.) columns) on AFHQ dataset with training and test noise level $\sigma_n = \sigma_t = 0.075$.

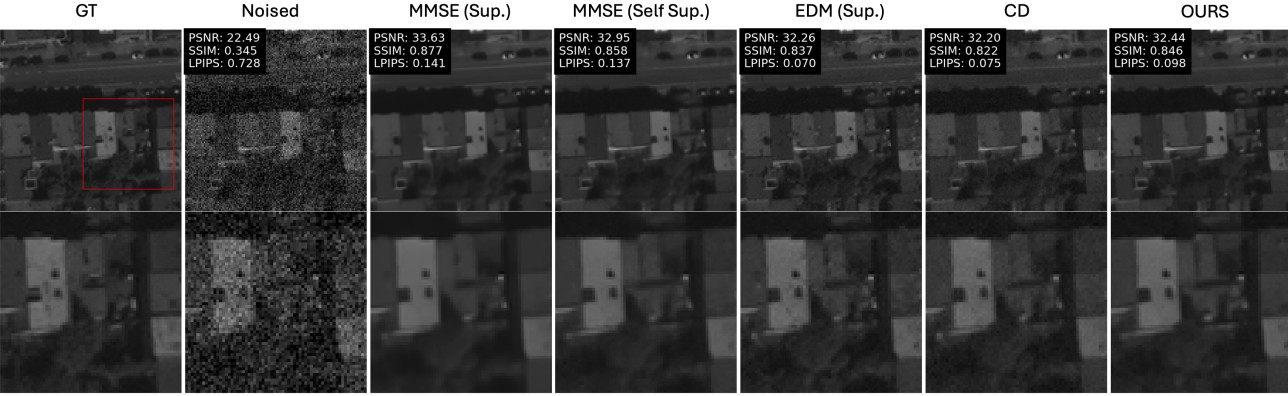

*Figure 9.* Example of various denoisers using diffusion sampling (except MMSE(Self Sup.) and MMSE(Sup.) columns) on NBU dataset with training and test noise level $\sigma_n = \sigma_t = 0.075$.

### A.4. Inference Procedure

For diffusion sampling in our experiments we use a slightly modified version of the samplers proposed in (Karras et al., 2022) by conducting sampling in measurement space. We show the inference procedure in Algorithm 1.

### A.5. Patch norms

To investigate the assumption that many real image distributions are approximately scale-invariant we plot the histogram of patch-wise norms for several image distributions using various patch sizes in Figure 14. We see that, in fact, we have a

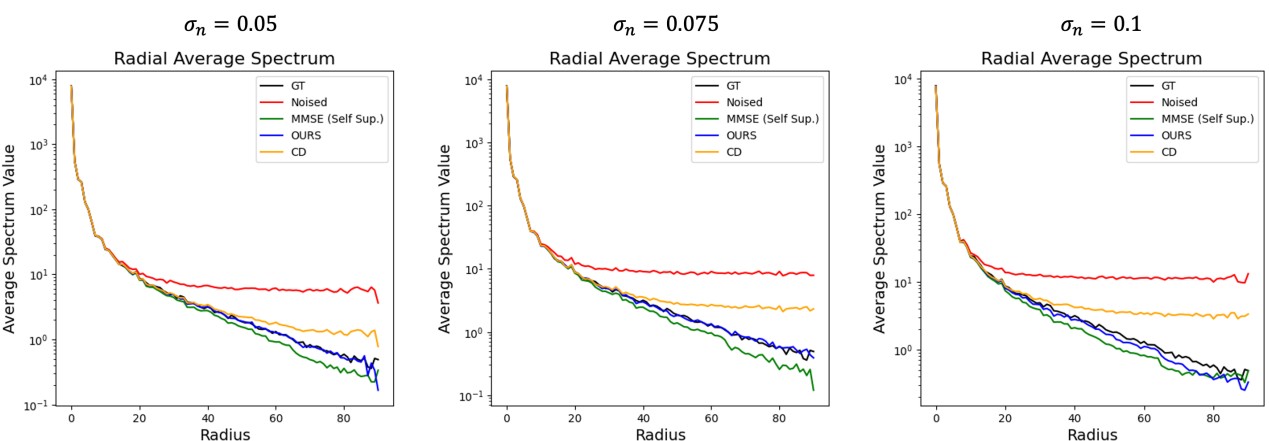

*Figure 10.* Example reconstructions for different training + inference noise levels on AFHQ.

*Figure 11.* Radial Spectrum of images in Figure 10.

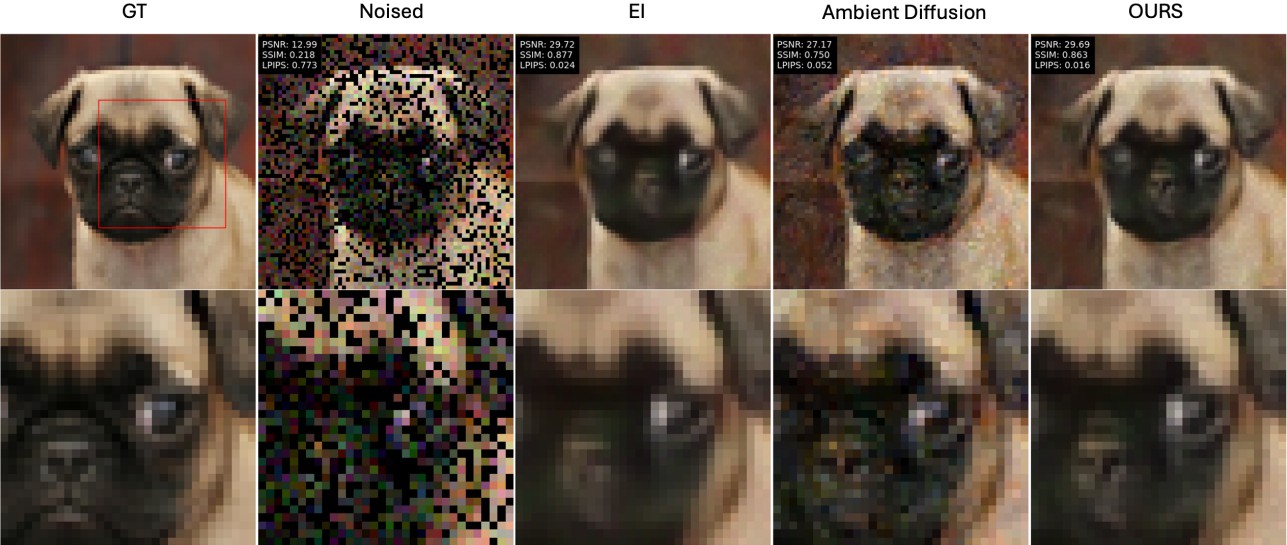

Figure 12. AFHQ inpainting example where the models are all trained in a self-supervised fashion.

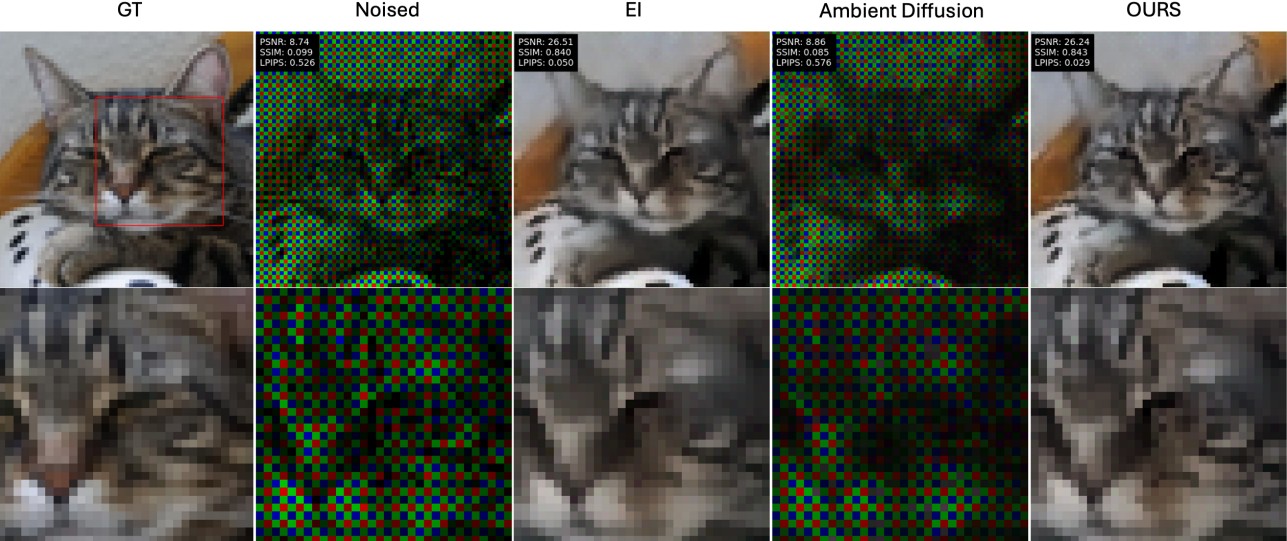

Figure 13. AFHQ demosaic example where the models are all trained in a self-supervised fashion.

spread in energy within each dataset which implies we may be observing a dataset that exhibits weak invariance.

### A.6. Extension to Blind Denoising

Our method can be straightforwardly extended to blind denoising problems ($\sigma_n$ is unknown) by appropriately modifying the SURE loss (4). To motivate this extension, we first illustrate how the performance of the method deteriorates if $\sigma_n$ is assumed known but specified incorrectly. This ablation is presented on the AFHQ denoising task with $\sigma_n = 0.075$, where we set $\sigma_n$ correctly ($\sigma_{exp} = 0.075$) and incorrectly ($\sigma_{exp} = 0.01, 0.05, 0.08, 0.10, 0.15$), See Figure 15. Note that the performance of self-supervised denoisers trained in a non-blind manner is highly sensitive to the correct specification of the noise level. Similar observations are reported in (Tachella et al., 2025a).

To extend our method to blind denoising settings, we leverage the UNSURE loss (Tachella et al., 2025a) in combination with our proposed loss giving the following training objective:

$$\min_{\theta} \max_{\sigma_n} \mathbb{E}_{\alpha,\mu} \left\{ \|\alpha \mathbf{y} + \mu \mathbf{1} - \mathrm{D}_{\boldsymbol{\theta}}(\alpha \mathbf{y} + \mu \mathbf{1}, \alpha \sigma_n)\|^2 + 2(\alpha \sigma_n)^2 \operatorname{div} \mathrm{D}_{\boldsymbol{\theta}}(\alpha \mathbf{y} + \mu \mathbf{1}, \alpha \sigma_n) \right\}, \qquad (13)$$

**Algorithm 1** Equivariant Sampling Inference

**Require:** $D_{\boldsymbol{\theta}}(\cdot, \sigma)$, $\{\sigma_K = \sigma_n, \ldots, \sigma_1 = \sigma_{\min}\}$, $\mathbf{y}$

1: $\mathbf{y}_{\text{next}} = \mathbf{y}$
2: **for** $i \in \{K, \ldots, 1\}$ **do**
3:      $\mathbf{y}_{\text{cur}} = \mathbf{y}_{\text{next}}$
4:      $\hat{\mathbf{x}} = \left(\mathbf{x}_{\text{cur}} - D_{\boldsymbol{\theta}}(\mathbf{A}^\top \mathbf{y}_{\text{cur}}, \sigma_i)\right)/\sigma_i$
5:      $\mathbf{y}_{\text{next}} = \mathbf{y}_{\text{cur}} + 2(\sigma_{i+1} - \sigma_i)\mathbf{A}\hat{\mathbf{x}}$
6:      $\boldsymbol{\eta} \sim \mathcal{N}(\mathbf{0}, \mathbf{I})$
7:      $\mathbf{y}_{\text{next}} = \mathbf{y}_{\text{next}} + \sqrt{2(\sigma_{i+1} - \sigma_i)\sigma_i}\,\boldsymbol{\eta}$
8:      **if** $i > 1$ **then**
9:          $\hat{\mathbf{x}}' = (\mathbf{x}_{\text{next}} - D_{\boldsymbol{\theta}}(\mathbf{A}^\top \mathbf{y}_{\text{next}}, \sigma_{i+1}))/\sigma_{i+1}$
10:          $\mathbf{y}_{\text{next}} = \mathbf{y}_{\text{cur}} + \frac{1}{2}(\sigma_{i+1} - \sigma_i)\mathbf{A}(\hat{\mathbf{x}} + \hat{\mathbf{x}}')$
11: **return** $\hat{\mathbf{x}}$

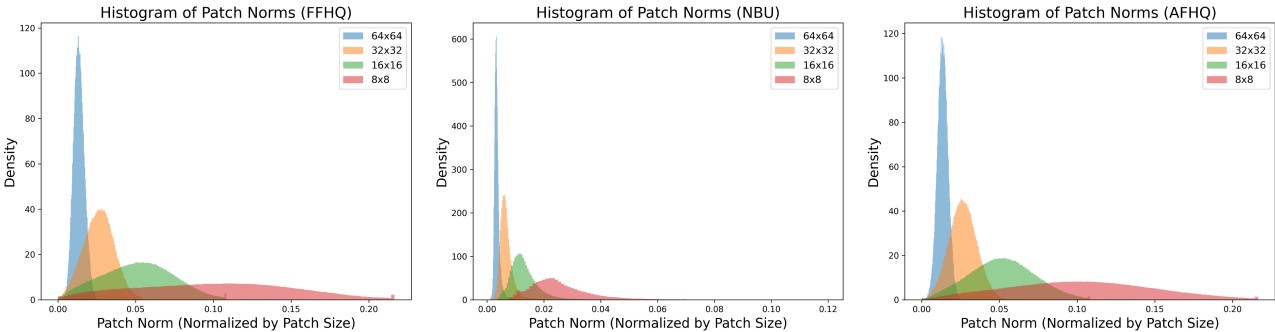

*Figure 14.* Histogram of image patches for each image distribution.

We trained a denoiser on the same AFHQ $\sigma_n = 0.075$ task except we initialize our estimate for $\sigma_n = 0.01$ and treat it as a learnable parameter in Equation (13). The resulting denoiser performance is shown in Figure 16. Note that using Equation (13) improves performance considerably across noise levels relative to an incorrect specification of the noise level.

### A.7. Training Schemes

To investigate how sampling schemes, at training time, on $\alpha$ and $\mu$ affect the denoiser performance we compared four models: **(1)** original denoiser where $\alpha, \mu \sim U$ (uniform) **(2)** denoiser where $ln(\sigma_n\alpha) \sim \mathcal{N}(-1.2, 1.2^2)$ which was proposed for training diffusion models in (Karras et al., 2022) (edm) **(3)** denoiser where $\alpha \sim U$ and $\mu = 0$ is fixed at training time ($\mu = 0$) **(4)** a denoiser where $\alpha \sim U(0.5, 1)$ instead of $U(0, 1)$ (narrow). We show the results in Figure 17. Here we see the sampling procedure for $\alpha$ does not seem to affect the performance of the model as long as we cover the full range of noise levels we wish to test on. We also see that not using $\mu$ leads to degraded performance as the noise level decreases.

### A.8. Inference Noise Scheduling

Here we provide a look at how $\sigma_t$ schedules and the number of inference steps (K) affects the performance of our denoiser. We test our method with $K = 5, 25, 100$ steps using three different schedules for $\sigma_t$. Specifically, we use alternative schedules for $\sigma_t$ by varying $\gamma$ in Equation (12). We tested performance at each step count with $\gamma = 0.5, 1, 7$. See Figure 18 examples of the noise schedules at $K = 5, 25, 100$. Figure 19 shows plots of performance with varying step counts $K$ and noise schedules $\sigma_t$ using our trained denoiser on the AFHQ dataset with $\sigma_n = 0.075$. We found that $\gamma = 7$ provided the best performance in the fewest number of steps and only minimal improvements are obtained with more than 25 inference steps using $\gamma = 7$. Our findings are consistent with prior work (Karras et al., 2022) which we based our inference noise schedule on.

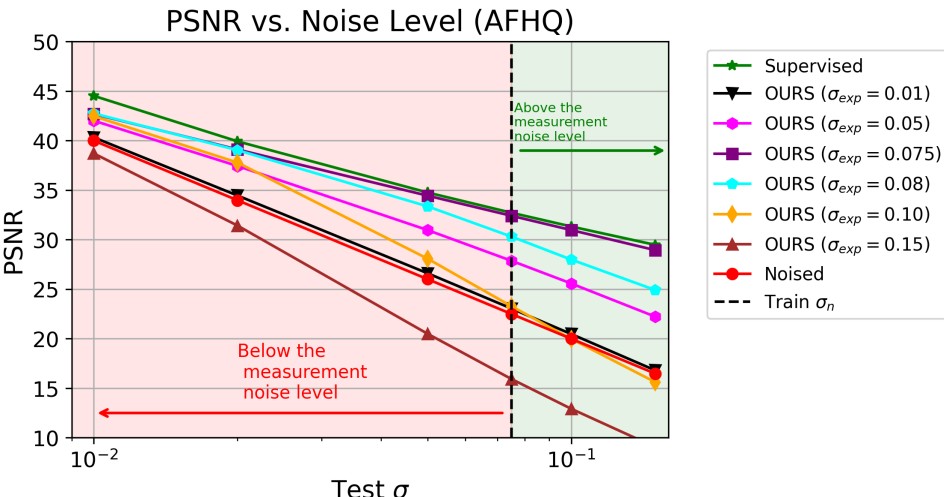

*Figure 15.* Denoising performance when the incorrect training noise level is used.

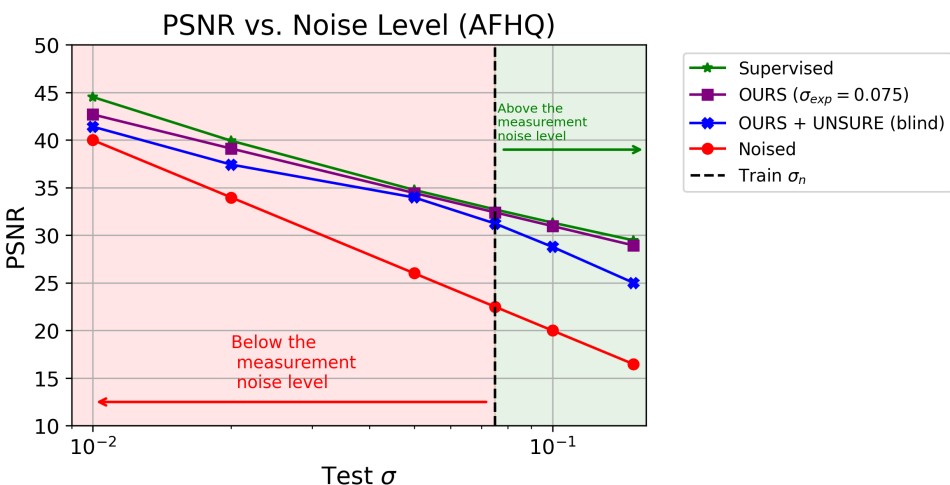

*Figure 16.* Denoising performance with UNSURE inspired loss in Equation (13) compared to our method using the true $\sigma_n$.

### A.9. MRI Reconstruction

To show that our method can generalize to other linear inverse problems we trained a model for MRI reconstruction. Here the data is complex valued and under-sampled using a 2D fourier transform followed by a single under-sampling mask and additive gaussian measurement noise ($\sigma_n = 0.1$). We use single coil brain data from the fastMRI dataset (Zbontar et al., 2019). Our training dataset consisted of $20,000$ noisy measurements. We chose rotations to use as the group of transformation to enforce equivariance for both our method and EI. Numerical results across a test set of 700 samples using our method compared to EI are shown in Table 3. An example reconstruction is presented in Figure 20 where we see better perceptual quality in the image contrast when comparing our technique to EI.

### A.10. Equivariance through Architecture

To investigate the performance of equivariant architectures in self-supervised settings we trained a normalization-equivariant architecture (Herbreteau et al., 2024) with SURE. We compared this method on the same AFHQ $\sigma_n = 0.075$ dataset used previously. Figure 21 shows how our denoiser, and those in the main paper, compare to using an equivariant denoising architecture. We see that although the equivariant architecture does help marginally at lower noise levels the performance drop is much greater than our approach. If we look at the example denoising images in Figure 22 we see that the equivariant

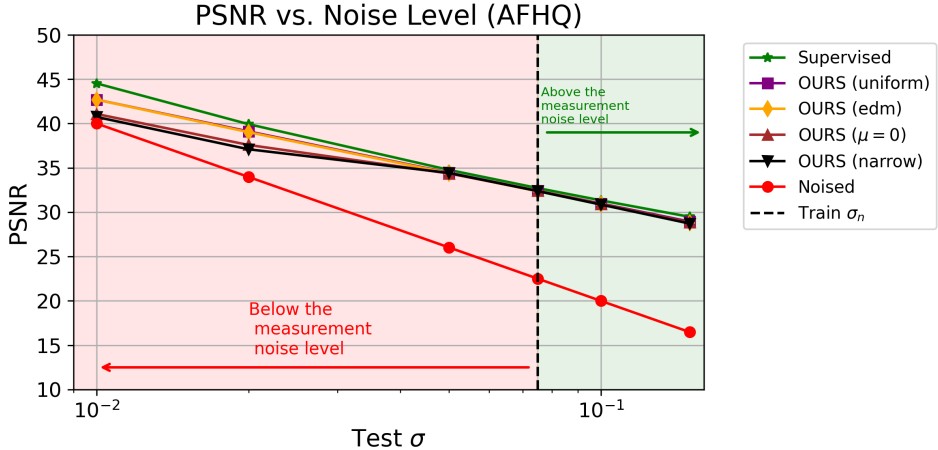

*Figure 17.* Denoising performance between models trained using different sampling procedures for $\alpha$ and $\mu$.

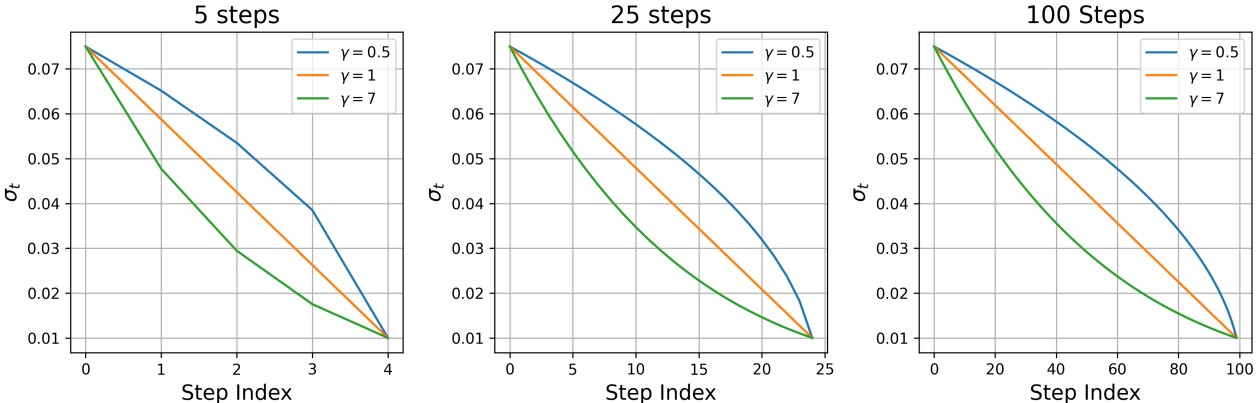

*Figure 18.* Example noise schedules for $\gamma = 0.5, 1, 7$.

architecture heavily over smooths the lower noise corrupted images compared to our method.

### A.11. Prospective low-field MRI Denoising

To further demonstrate that our method works on real sensor data we applied our method to an in-house MRI dataset of 62 subjects whose wrists were scanned on a 1-Tesla permanent magnet MRI scanner with Institutional Review Board approval and informed consent. The scans were fully sampled using a multi-echo spin-echo (MESE) sequence, and thus the inherent SNR in each sample variable, with measured noise levels $\sigma_n \in [0.04, 0.06]$. Our training set consisted of 30,000 noisy images (see Figure 23). The noise level was estimated for each image in the training dataset by sampling background pixels (top left or bottom right patch) and calculating the standard deviation. Our denoiser was trained directly on the inverse fourier transform of the raw measurements. A self-supervised denoising example using our method is shown in Figure 24. Since all the measurements are corrupted with noise in this real dataset we cannot provide image quality metrics. However, we visually observe that our method is removing measurement noise and can provide both a plausible sample along with a

| Task | $\sigma_n$ | Solver | Sampler | Self Sup. | PSNR (↑) | SSIM (↑) | LPIPS (↓) | FID (↓) |
|---|---|---|---|---|---|---|---|---|
| MRI Recon. | 0.075 | EI | | ✓ | **23.16** | 0.796 | 0.043 | 86.94 |
| | | OURS | ✓ | ✓ | 22.22 | **0.812** | **0.027** | **72.88** |

*Table 3.* Fourier under-sampling on single coil fastMRI $64 \times 64$.

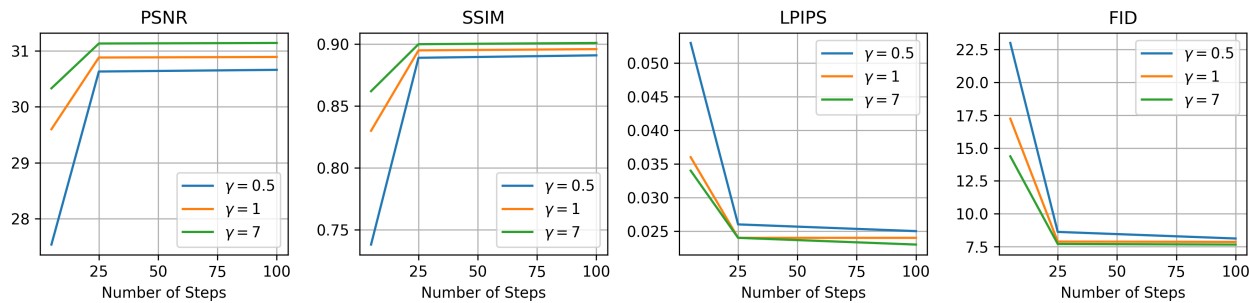

*Figure 19.* Sampling metrics using various $\gamma = 0.5, 1, 7$ and step counts $K = 5, 25, 100$ in Equation (12).

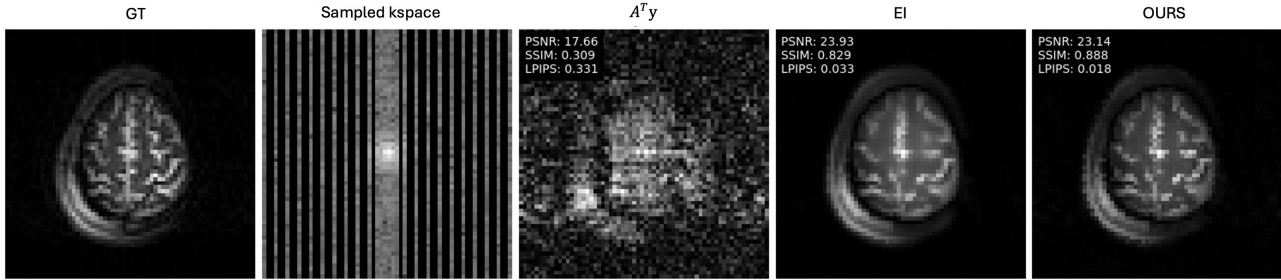

*Figure 20.* MRI reconstruction example.

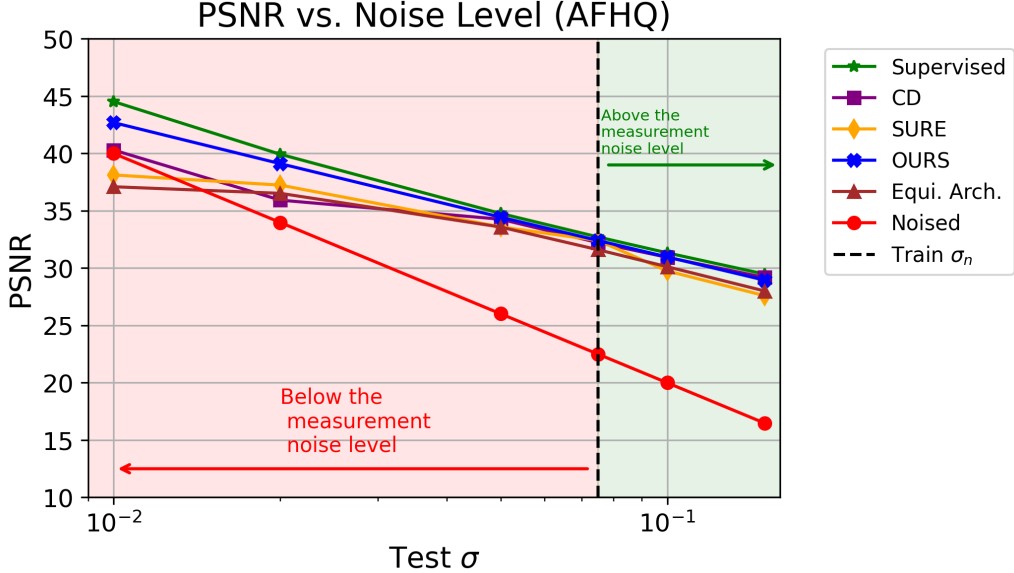

*Figure 21.* Denoiser comparison on AFHQ $\sigma_n = 0.075$ denoising task.

variance map calculated from sampling 10 different reconstructions.

## A.12. Complete proof of Theorem 3.1

*Proof.* Let $\mathcal{P}(\mathbb{R}^n)$ be the class of functions on $\mathbb{R}^n$ that are positively homogeneous and whose logarithm is twice continuously differentiable with Lipschitz continuous gradient. First, assume that $p(\mathbf{x})$ admits a factorization $p(\mathbf{x}) = p_1(\mathbf{x})p_2(\mathbf{x})$, with $p_1(\mathbf{x})$ and $p_2(\mathbf{x})$ depending only on $\|\mathbf{x}\|$ and $\mathbf{x}/\|\mathbf{x}\|$ respectively, and let $\tilde{p}_1$ be the function in $\mathcal{P}(\mathbb{R}^n)$ that is closest to $p_1$ in

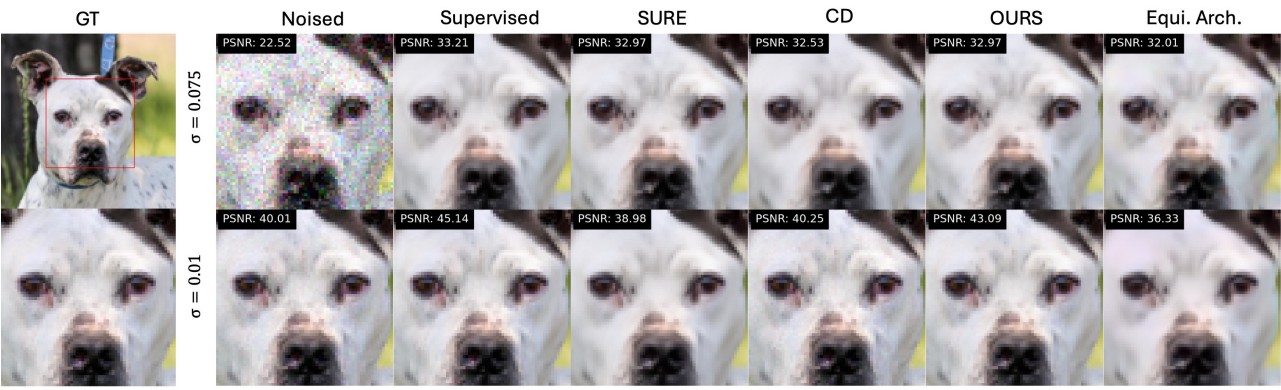

*Figure 22.* Example Denoiser comparison on AFHQ denoising task for inference at the training noise level $\sigma = 0.075$ and below the training noise level $\sigma = 0.01$.

the sense of the Fisher divergence w.r.t. the posterior $p(\mathbf{x}|\mathbf{y}, \sigma) \propto p(\mathbf{x})p(\mathbf{y}|\mathbf{x}, \sigma)$, i.e.,

$$\tilde{p}_1 = \underset{q \in \mathcal{P}(\mathbb{R}^n)}{\arg\min} \int_{\mathbb{R}^n} \|\nabla \log q(\mathbf{x}) - \nabla \log p_1(\mathbf{x})\|^2 p(\mathbf{x}|\mathbf{y}, \sigma)\mathrm{d}\mathbf{x}$$

Consider the approximation $\tilde{p}(\mathbf{x}) \propto \tilde{p}_1(\mathbf{x})p_2(\mathbf{x})$ of $p$, obtained by replacing the correct marginal $p_1$ by $\tilde{p}_1$, and denote by $\tilde{p}(\mathbf{x}|\mathbf{y}, \sigma) \propto \tilde{p}(\mathbf{x})p(\mathbf{y}|\mathbf{x}, \sigma)$ the associated posterior distribution. We view $\tilde{p}$ as an operational prior that may be improper, but we assume that $\tilde{p}(\mathbf{x}|\mathbf{y}, \sigma)$ is well defined. Moreover, we denote by $\kappa_\sigma$ the Fisher divergence between the posteriors $p(\mathbf{x}|\mathbf{y}, \sigma)$ and $\tilde{p}(\mathbf{x}|\mathbf{y}, \sigma)$, given by

$$\kappa_\sigma = \int_{\mathbb{R}^d} \|\nabla \log p_1(\mathbf{x}) - \nabla \log \tilde{p}_1(\mathbf{x})\|^2 \pi(\mathbf{x}|\mathbf{y}, \sigma)d\mathbf{x},$$

where we have used the factorization property of $p$ and $\tilde{p}$ and Bayes' rule to simplify $\nabla \log p(\mathbf{x}|\mathbf{y}, \sigma) - \nabla \log \tilde{p}(\mathbf{x}|\mathbf{y}, \sigma)$.

Furthermore, because $\mathbf{x} \mapsto \log p(\mathbf{y}|\mathbf{x}, \sigma)$ is $1/\sigma^2$-strongly concave, and the Hessian of $\log \tilde{p}$ is bounded, there exists some $\sigma^\star$ such that for all $\sigma \leq \sigma^\star$ the approximation $\log \tilde{p}(\boldsymbol{x}|\boldsymbol{y}, \sigma)$ is strongly concave outside some compact set (i.e., for some constants $K > 0$ and $R \geq 0$, $\nabla^2 \log \tilde{p}(\mathbf{x}|\mathbf{y}) \succeq K\mathbf{I}$ for all $\|\mathbf{x}\| \geq R$). From (Huggins et al., 2018, Theorem 5.3), this implies that for any $\sigma \leq \sigma^\star$, the 2-Wasserstein distance between $p(\mathbf{x}|\mathbf{y}, \sigma)$ and $\tilde{p}(\mathbf{x}|\mathbf{y}, \sigma)$ is bounded as

$$\mathcal{W}_2\left(p(\mathbf{x}|\mathbf{y}, \sigma), \tilde{p}(\mathbf{x}|\mathbf{y}, \sigma)\right) \leq \psi \kappa_\sigma,$$

where $\psi > 0$ depends on $\tilde{p}(\mathbf{x}|\mathbf{y}, \sigma)$, but is independent of $p(\mathbf{x}|\mathbf{y}, \sigma)$. For example, when $\tilde{p}(\mathbf{x}|\mathbf{y}, \sigma)$ is strongly log-concave, $\psi$ is the inverse of the log-concavity constant.

Following on from this, we denote by $\tilde{\mathrm{D}}(\mathbf{y}, \sigma)$ the MMSE denoiser associated with $\tilde{p}(\mathbf{x}|\mathbf{y}, \sigma)$ and use Equation (11) to show that $\tilde{\mathrm{D}}(\mathbf{y}, \sigma)$ verifies the desired rescaling property $\tilde{\mathrm{D}}(\mathbf{y}, \sigma') = \frac{\sigma'}{\sigma}\tilde{\mathrm{D}}(\frac{\sigma}{\sigma'}\mathbf{y}, \sigma)$. Lastly, because $\mathcal{W}_2$ bounds the difference in the expectation of random variables, we have

$$\|\mathrm{D}(\mathbf{y}, \sigma') - \tilde{\mathrm{D}}(\mathbf{y}, \sigma')\| \leq \psi \kappa_{\sigma'},$$
$$\|\tfrac{\sigma'}{\sigma}\mathrm{D}(\tfrac{\sigma}{\sigma'}\mathbf{y}, \sigma) - \tfrac{\sigma'}{\sigma}\tilde{\mathrm{D}}(\tfrac{\sigma}{\sigma'}\mathbf{y}, \sigma)\| \leq \tfrac{\sigma'}{\sigma}\psi \kappa_\sigma,$$

and therefore,

$$\|\mathrm{D}(\mathbf{y}, \sigma') - \tfrac{\sigma'}{\sigma}\mathrm{D}(\tfrac{\sigma}{\sigma'}\mathbf{y}, \sigma)\| \leq \epsilon,$$

with $\epsilon^2 = \psi^2 \kappa_{\sigma'}^2 + (\frac{\sigma'}{\sigma})^2 \psi^2 \kappa_\sigma^2$, concluding the first part of the proof.

The argument is extended to to cases where $p(\mathbf{x})$ does not admit the desired factorization by introducing an intermediate approximation $\bar{p}(\mathbf{x})$ of $p(\mathbf{x})$ that satisfies the required approximation factorization and is chosen to minimize the 2-Wasserstein distance between $\bar{p}(\mathbf{x}|\mathbf{y}, \sigma) \propto p(\mathbf{y}|\mathbf{x}, \sigma)\bar{p}(\mathbf{x})$ and $p(\mathbf{x}|\mathbf{y}, \sigma) \propto p(\mathbf{y}|\mathbf{x}, \sigma)(\mathbf{x})$, which we denote by $\lambda_\sigma > 0$. We then apply the original argument to $\bar{p}(\mathbf{x}|\mathbf{y}, \sigma)$ instead of $p(\mathbf{x}|\mathbf{y}, \sigma)$, and use the triangle inequality to obtain

$$\mathcal{W}_2\left(p(\mathbf{x}|\mathbf{y}, \sigma), \tilde{p}(\mathbf{x}|\mathbf{y}, \sigma)\right) \leq \sqrt{\psi^2 \kappa_\sigma^2 + \lambda_\sigma^2},$$

where now $\kappa_\sigma$ is the Fisher divergence between the posteriors $\bar{p}(\mathbf{x}|\mathbf{y}, \sigma)$ and $\tilde{p}(\mathbf{x}|\mathbf{y}, \sigma)$. Again, because $\mathcal{W}_2$ bounds the difference in the expectation of random variables, we have

$$\|\mathrm{D}(\mathbf{y}, \sigma') - \tilde{\mathrm{D}}(\mathbf{y}, \sigma')\| \leq \sqrt{\psi^2 \kappa_{\sigma'}^2 + \lambda_{\sigma'}^2}\,,$$
$$\|\tfrac{\sigma'}{\sigma}\mathrm{D}(\tfrac{\sigma}{\sigma'}\mathbf{y}, \sigma) - \tfrac{\sigma'}{\sigma}\tilde{\mathrm{D}}(\tfrac{\sigma}{\sigma'}\mathbf{y}, \sigma)\| \leq \tfrac{\sigma'}{\sigma}\sqrt{\psi^2 \kappa_\sigma^2 + \lambda_\sigma^2}\,,$$

and therefore,

$$\|\mathrm{D}(\mathbf{y}, \sigma') - \tfrac{\sigma'}{\sigma}\mathrm{D}(\tfrac{\sigma}{\sigma'}\mathbf{y}, \sigma)\| \leq \epsilon'\,,$$

with $\epsilon'^2 = \psi^2 \kappa_{\sigma'}^2 + \lambda_{\sigma'}^2 + (\tfrac{\sigma'}{\sigma})^2 (\psi^2 \kappa_\sigma^2 + \lambda_\sigma^2)$, concluding the proof.

$\square$

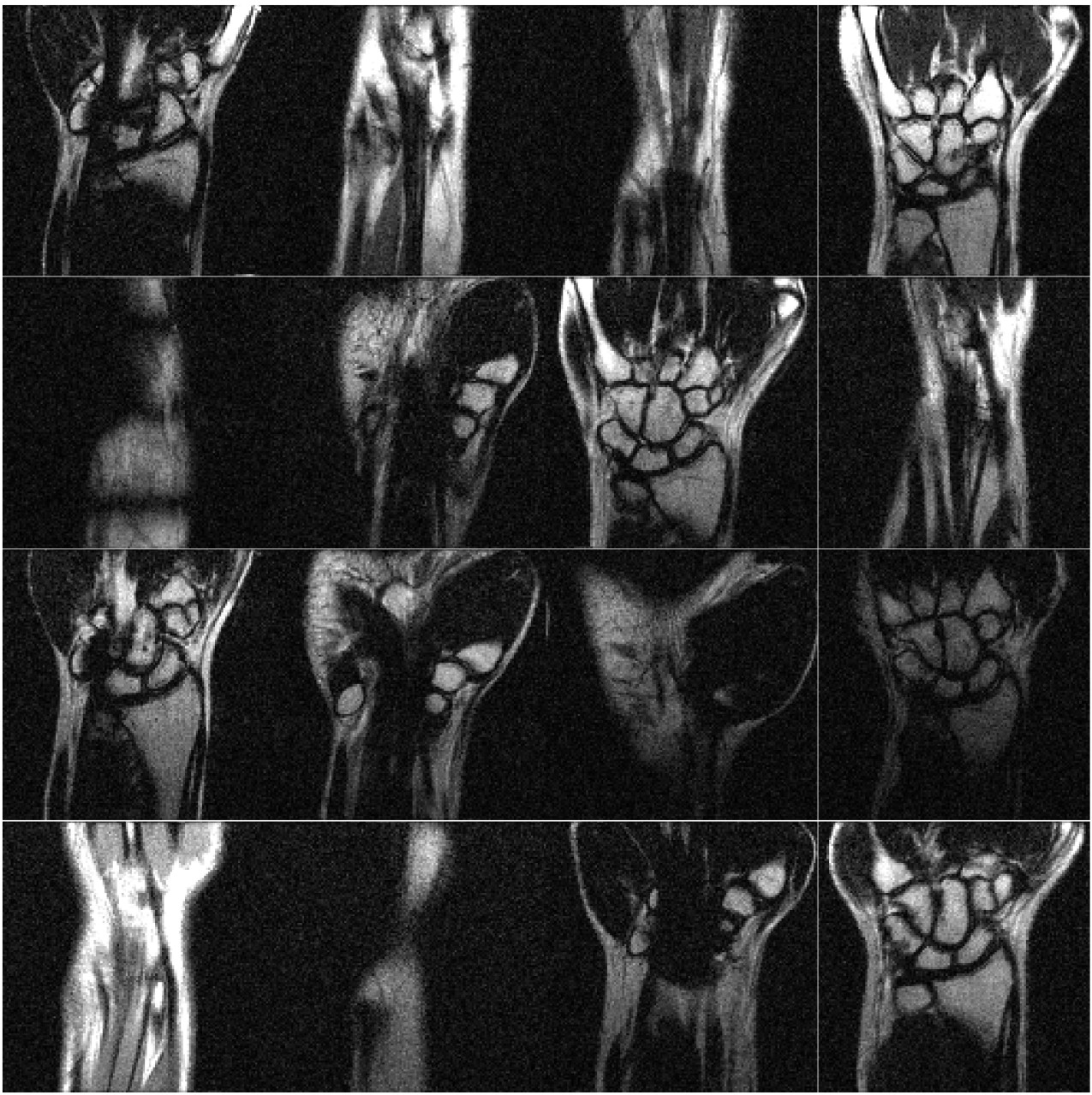

*Figure 23.* Prospectively collected MRI training data for self-supervised denoising experiments.

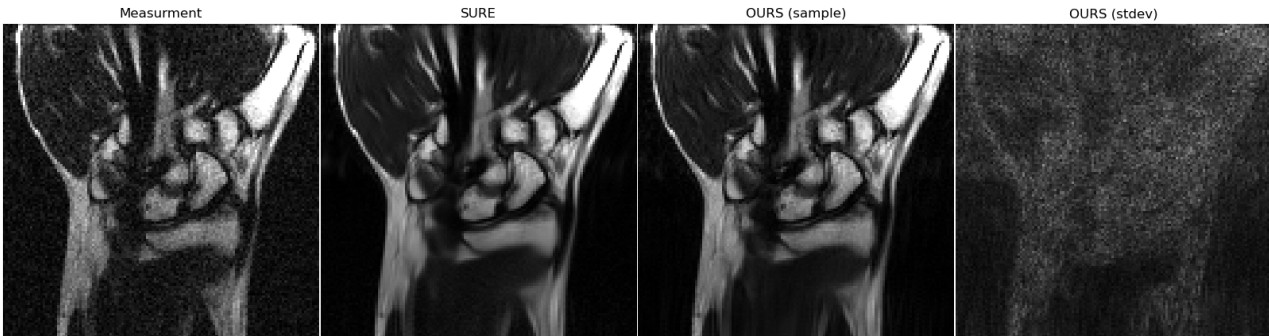

*Figure 24.* Prospectively collected MRI test data denoised using our self-supervised trained denoiser + sampler.

