# OpenReview forum: "Normalization-equivariant Diffusion Models: Learning Posterior Samplers From Noisy And Partial Measurements"
_ICML.cc/2026/Conference — ICML 2026 regular_

### Official Review · Reviewer_Q1pC · 2026-02-18

**Soundness:** 3
**Presentation:** 4
**Significance:** 3
**Originality:** 4
**Overall Recommendation:** 5
**Confidence:** 4

**Summary:**

The authors propose normalization-equivariant diffusion model as a method to train a diffusion model for posterior sampling when only the corrupted measurements are available. Prior methods such as SURE are known to recover MMSE estimates on the measurement noise level, but cannot act as posterior samplers as MMSE estimates below the measurement noise level cannot be obtained. CD partially overcomes this but is limited when there does not exist any clean data to assist the training process. The authors show that with the proposed NE-diffusion, they are able to achieve posterior sampling even when the diffusion model is trained solely on corrupted measurements by additionally enforcing normalization equivariance to the loss.

**Compliance With Llm Reviewing Policy:**

Affirmed.

**Key Questions For Authors:**

1. How does the proposed method compare against other diffusion posterior samplers that learns from corrupted measurements only? For instance, I am curious how it compares to GSURE-diffusion [1] and DiEM [2] since both tackle a similar problem, and additionally possibly DDIP [3].


2. I understand that the experiments focus on posterior sampling, but I am also curious how the performance is for sampling from the prior after model training.

**Limitations:**

Yes

**Strengths And Weaknesses:**

### Strenghts

1. Learning only from corrupted measurements is one of the most important yet-to-be-solved problems in inverse problems. This work is a valuable contribution to this field.

2. Normalization equivariance is a straightforward and sound regularization, and the authors back it up with some theoretical arguments.

3. The experiments are complete. The authors not only experiment with Gaussian denoising, but extends to linear inverse problems, and even to real-world measurements.

4. The paper is concise and well-written.

5. The method reports strong performance.


### Weaknesses

1. The experiments/discussion could be improved with more relevant baselines, e.g. [1-3]. See Key questions.

2. The authors state immediately after (11) that positively homogeneous condition cannot be met. Also, the assumption of theorem 3.1. is relatively strong.


### References

[1] Kawar, Bahjat, et al. "Gsure-based diffusion model training with corrupted data." TMLR 2024.

[2] Rozet, François, et al. "Learning diffusion priors from observations by expectation maximization." NeurIPS 2024.

[3] Chung, Hyungjin, and Jong Chul Ye. "Deep diffusion image prior for efficient ood adaptation in 3d inverse problems." ECCV 2024.

---

> ### Author Rebuttal · Authors · 2026-03-31
>
> We thank the reviewer for their time and feedback on our paper.
>
> (1) Additional method comparisons
>
> The GSURE method requires measurements with a small noise level associated to varying/randomized measurement operators (e.g., randomly missing pixels that change from image to image), whereas we tackle the setting of incomplete data from a single operator. In the case of denoising, the GSURE approach is equivalent to the SURE method reported in our experiments (see Figs. 1 and 2), which fails to learn below the measurement noise level $\sigma_n$.
>
> We do not compare with DiEM, as we found that the method required prohibitively long training times to obtain good results, since the diffusion model needs to be retrained multiple times within an iterative expectation-maximization framework. In contrast, our method requires only a single additional denoiser evaluation per mini-batch compared to the standard supervised setting.
>
> We also do not compare with the DDIP method by Chung et al., which requires a pre-trained denoiser (that is then fine-tuned to handle out-of-distribution data), whereas our method can train a denoiser from random weights in a fully self-supervised way.
>
> (2) Strength of assumption for Theorem 3.1
>
>  The derivation in Line 145 shows that Gaussian MMSE denoisers are exactly normalization equivariant when the prior is invariant to changes in scale, i.e., $p(\alpha x) = p(x)$ for all $x$ and all $\alpha \in \mathbb{R}_{+}$. This condition cannot hold in practice, as it violates the fundamental requirement that $\int p(x) \textrm{d}x =1$.
>
> However, Theorem 3.1 establishes that Gaussian MMSE denoisers are approximately equivariant under conditions tied to structural properties of the prior that, although seemingly strong, are in fact widely observed in natural signals and hence can be regarded as mild and natural. In particular, Theorem 3.1 holds when $p(x) \approx p_1(\|x\|)p_2(\frac{x}{\|x\|})$, where the first term depends only on the scale of the signal and is locally polynomial, for example, when the signal scale follows a heavy-tailed distribution. This property is well documented in natural image datasets and is exploited in practice, for instance by the DRUNet denoiser [1], which achieves state-of-the-art performance. For completeness, Figure 14 in the Appendix illustrates this behavior.
>
> Theorem 3.1 is significant because it indicates that accurate surrogates for the MMSE denoiser can be learned in a fully self-supervised manner across a broad range of noise levels below the measurement noise. Figure 1 provides strong empirical support for this claim.
>
> We will clarify these points further in the final version of the paper.
>
>
> (3)Prior Sampling
>
>  Our method can be straightforwardly adapted for prior sampling via the following minor modification. In our particular training scheme, we only trained our diffusion models for $0<\sigma_t \leq 2\sigma_n$ since we were considering posterior sampling. In prior sampling, where one would need to start sampling from a purely noisy sample, we would need to train the denoiser to cover noise levels higher than $2\sigma_n$. This can be easily achieved by adding additional noise to the already noisy signals during training, and applying SURE (see equation in response 3 to reviewer Q1pc).
>
> [1] Plug-and-Play Image Restoration With Deep Denoiser Prior

---

> > ### Author Rebuttal · Reviewer_Q1pC · 2026-04-01
> >
> > My concerns have been addressed.

---

> > > ### Author Response · Authors · 2026-04-03
> > >
> > > Thank you for acknowledging our rebuttal and letting us know that your comments are fully addressed. We kindly ask that you reconsider your final score to reflect your agreement with the discussion and changes we made

---

### Official Review · Reviewer_5Mgr · 2026-03-10

**Soundness:** 3
**Presentation:** 4
**Significance:** 3
**Originality:** 4
**Overall Recommendation:** 4
**Confidence:** 4

**Summary:**

This paper works on the problem of training an image restoration estimator when we can only access the degraded images. This is an important problem in diffusion models for image restoration. The authors extend over the Stein's risk estimator framework (SURE), by allowing for arbitary measurement noise level and diffusion noise scheduling. The paper has both theoretical and empirical results, and the results show that the proposed method improve upon SURE.

**Compliance With Llm Reviewing Policy:**

Affirmed.

**Final Justification:**

The authors have adderssed most of my concerns. My only concern is the impact of the proposed method in general. The effectiveness for y = Ax+eps problems have only been confirmed on one experiment (i.e., MRI).

**Key Questions For Authors:**

# Important

- Equation 4. I don't understand why you have $D_\theta(y, \sigma_n)$ stead of $D_\theta(y, \sigma_t)$. Note that the MSE estimator is $x \mapsto D_\theta(x, \sigma_t)$ in Equation 1, hence, SURE should apply to $y \mapsto D_\theta(y, \sigma_t)$. The correct equation should be
$$
\lVert y - D_\theta(y, \sigma_t)^2 + 2 \sigma_n D_\theta(y, \sigma_t)\rVert.
$$
As such, I also don't understand "A key limitation of SURE is that it can only be used to obtain an unbiased estimate of the supervised denoising loss ..." However is the diffusion noise scheduler related to the measurement noise in the SURE framework?
- I agree with SURE can be applied to measurement $y = x + \sigma_n \eta$, however, I disagree if it can apply to $y = A x + \sigma_n \eta$ with some operator $A$.
The former gives you an unbiased estimate to the MSE, yes, but, this is not true with $A$. The SURE esimator for $y = A x + \sigma_n \eta$ is unbiased for the **modified** MSE ||Ax - A D_\theta(x, \sigma)|| which does not correspond to (1).
- I don't follow how you implement the normalization equivariance property in the denoiser. Usually, the denoiser is a neural network that takes $x$ and $t$ as inputs, not $x$ and $\sigma_t$. As such, how do you guarantee the property over $t$?
- Can you explain the role of $\mu$?. The authors claim "we find that this leads to better performance for self-supervised learning", but I can't believe if this is true. Fundamentally, $+ \mu 1, \quad \mu \sim U(0, 1)$ means that we are even more corrupting the measurement $y$, and it should be harder to recover the truth.
- Line 145. What are the prior and the assumption? You have assumed the equivariance property on the neural network denoiser, how is it related to the true prior?
- Theorem 3.1, "... p(x) = p1(x)p2(x), with p1(x) and p2(x) depending only on |x| and x/|x| respectively ...". It is hard to understand the meaning of it. Is this an assumption introduced only to facilitate the proof, or does it have a meaning? Can it be verified in practice? Can you give one example? Also, what does it mean by "x/|x| is **independent** of |x|"?
- Theorem 3.1. I think the theorem can be improved further. The result tells that there exits a $\sigma^\star$, and I think it also depends on $\epsilon$. It is not clear how tight the interval $(0, \sigma^\star)$ is, which is crucial for choosing this hyperparameter in practice.
- Figure 1 is a bit counter-intuitive. As the test noise $\sigma$ increases, the difference between "supervised" and "SURE and variants" should increase, and vice versa. As such, the yellow line make sense, but not for the others. Why is the proposed method, which still operates under the SURE framework, is still in line with the supervised approach?
Think also in another way: The SURE estimator of the loss function can only be worse than the original supervised loss, by construction.
- In Tables 1 and 2, it is not clear whether "OURS" outperforms.

# Minor

The abstract is really too long. It is hard to digest the take-away message from it.

Equation 1, first line, $x$ is missing the subscript $i$. There should also be an integral over $t$ in the loss function.

It is confusing that (5) has a square over $\sigma_n$ but (3) doesn't.

**Limitations:**

No. The authors have talked about future work briefly in Conclusion but these are not limitation per se.

**Strengths And Weaknesses:**

Soundness: Overal sound. There are a few typos, but can be fixed. I have some questions of the theorem. See Comments.

Presentation: The presentation is clear and easy to follow.

Significance: I think the developed method is a good increment to the community, in the sense that, the new method improves upon the baseline SURE, but is not overly complicated. Simple method but works great. The limitation is the flexibility of the neural network parameterization. However, the paper does not show significant improvement for y = Ax + eps problems.

Originally: Yes, it provides an improved understanding of applying SURE for diffusion image restoration.

---

> ### Author Rebuttal · Authors · 2026-03-31
>
> We greatly appreciate the in-depth comments/questions the reviewer has provided.
>
> (1) Limitation in parameterization of architecture
>
> We would like to clarify that our method does not explicitly enforce the architecture parameterization to be normalization equivariant, but rather, we enforce equivariance through the loss function in Equation 8. We refer to (4) in our response to 2n5Z.
>
> (2) Limited improvement in non-trivial $A$ case
>
> We acknowledge that in the case where $A\neq I$, the improvements of our method over existing approaches are not as impressive. We would, however, like to point the reviewer's attention to our example with MRI reconstruction results in Appendix section A.9 (Fig. 20, Table 3) where we do see clear improvements. We believe this is because the SNR in this case was lower than the other experiments, which is where our method is meant to outperform point estimators like EI.
>
>
> (3) Clarifying Equation 4
>
> We apologize for the confusion related to Eq. 4 - our presentation was not sufficiently clear. In Eq. 4, we introduce the SURE objective for the case where a denoiser is trained only at the measurement noise level $\sigma_n$. For the case where $\sigma_t=\sigma_n$ in the diffusion schedule, this self-supervised denoiser is (in expectation) an unbiased estimator of the supervised case. However, learning a diffusion model requires learning the MMSE estimator for all $\sigma_t$. Learning the MMSE estimator with SURE above the noise level of the data (i.e., $\sigma_t>\sigma_n$) can be done with the corrupted measurements ($y=x+ \sigma_n\eta$) by simply adding more noise to the measurements:
>
> $\mathcal{L}_{\text{SURE}}(y,\theta) = || y' - D(y', \sigma_t )||_2^2 + 2\sigma_t ^{2}div D(y',\sigma_t )$
>
> where $y' = y+\sqrt{\sigma_t^2-\sigma_n^2}\epsilon$ and $\epsilon \sim \mathcal{N}(0,I)$. This approach can be used for noise levels $\sigma_t > \sigma_n$, but not $\sigma_t<\sigma_n$.  We will clarify this point in the updated paper.
>
> (4) Applying SURE to $y = Ax + \sigma_n\eta$
>
> Indeed, as the reviewer points out, in the case of $y = Ax + \sigma_n\eta$, SURE does not provide an unbiased estimate of the MSE in the pixel space. However, SURE does provide an unbiased estimator of the MSE in the measurement space of $y$, that is it is an unbiased estimator of the clean measurement consistency $E_{y,x}||A\operatorname{D}_{\theta}(y,\sigma_n) - Ax||^2_2$
>
> If $A$ is invertible, this loss has the same global minimum as the MSE in pixel space. If $A$ is not invertible (the case we consider in Section 5), the loss is not an estimator of the MSE loss. Thus, we add the equivariant imaging (EI) loss term to approximate the supervised MSE los
>
> (5) Connecting $t$ and $\sigma_t$
>
> In our work, we follow the scheme from Karras (2022) where $\sigma_t = t$.
>
> (6) The role of $\mu$
>
> The motivation behind adding $\boldsymbol{1}\mu$ is to learn a normalization equivariant denoiser, going beyond scale invariance enforced by $\alpha$. The mean shift term $\boldsymbol{1}\mu$ in Eq. 8 is added to both the input of the network and the target variable, and is thus not additional noise that the network has to remove. This means we do not need to estimate a further corrupted signal, but rather estimate the shifted clean signal from the shifted corrupted measurement. We emphasize that we know the identity of $\mu$ at each training batch just as we know $\alpha$.
>
>
> (7) Clarifying Theorem 3.1 and Line 145
>
>  Due to space constraints we ask the reviewer to see our response (2) to reviewer (Q1pc).
>
> (8) Clarifying Figure 1
>
> In the case of our method and CD, variants of the loss function in response (3) are used to learn self-supervised denoisers for $\sigma_t > \sigma_n$. Leading to competitive performance at $\sigma_t > \sigma_n$. In the case of SURE (yellow line) in Figure 1, we only train this denoiser at a single noise level $\sigma_t = \sigma_n$ to show how self-supervised denoisers degrade at test time (even for $\sigma_t > \sigma_n$) if not properly handled. It is therefore expected that the yellow line diverges from the supervised performance above and below the measurement noise level while other self-supervised methods do not.
>
> (9) Clarifying Tables 1 and 2
>
> Tables 1 and 2 could certainly be more clear. We have both sup. and self-sup. approaches in the tables. Our method should be mainly compared to the other self-sup. techniques (i.e., MMSE (self-sup), CD, EI) and the supervised rows should be consider the "gold-standard" for their given estimator class (i.e., for generative (Sampler) and point (Non-sampler) estimates). With this distinction, our method outperforms all other self-supervised methods in perceptual metrics (FID, LPIPs) which was the goal of our work. In the final paper we will make this distinction clearer in the tables.
>
> (10) Minor comments
>
> In the final paper, we will 1) parse down the abstract, 2) fix equation 1, and 3) remove the square over $\sigma_n$ in Equation 5.

---

> > ### Author Rebuttal · Reviewer_5Mgr · 2026-04-02
> >
> > Thank you for your replies!
> >
> > I agree that your results for MRI reconstruction is impressive, but my concerns on y = Ax+eps in general was not addressed.
> >
> > Theorem. Thank you for pointing me to the replies to Q1pc. Indeed you have answered to the reviewer on the strength of the theorem, but my questions were more specific, and several were not addressed, let me repeat:
> >
> > - Can it be verified in practice? Can you give one **constructive** example? Also, what does it mean by "x/|x| is independent of |x|"?
> > - I think the theorem can be improved further. The result tells that there exits a $\sigma^\star$ ...
> > - I checked the paper [1] you provided. But I do not see how your statement "This property is well documented in natural image datasets and is exploited in practice" is supported in that paper. Can you pinpoint to me where I can find the support in [1]?

---

> > > ### Author Response · Authors · 2026-04-04
> > >
> > > Thank you for acknowledging our rebuttal and for your follow-up questions.
> > >
> > > - Independence of x/|x| from |x| implies that knowing the norm |x| conveys no information about the individual pixel values that contribute to it. In polar coordinates, this corresponds to the radial component being independent of the angular component, the latter encoding the normalized pixel values. Since |x| sums information across all pixels, the individual pixels in the normalized image x/|x| become asymptotically independent of |x| as the dimension increases, provided the pixel dependencies are sufficiently weak (i.e., adding more pixels increases the total information content of the image). This phenomenon has been studied extensively in several settings, e.g., high-dimensional log-concave distributions (see https://doi.org/10.1214/10-AOP592) and vectors with independent components (see https://arxiv.org/abs/1311.0587); and it holds exactly for isotropic Gaussian vectors in any dimension.
> > >
> > > Developing estimators that can assess the validity of these assumptions directly from the measurements y, particularly in the context of self-supervised training, is an interesting direction for future work. To the best of our knowledge, no such techniques are currently available.
> > >
> > > - Regarding improving the main theorem: We agree that the theorem could be strengthened to yield a sharper characterization of the range of \sigma for which the bound holds, as well as more explicit quantitative control over \epsilon. However, to the best of our knowledge, this would require substantially stronger assumptions on p(x) that would be difficult to check in practice and may therefore limit the practical relevance of the result.
> > >
> > > Importantly, Figure 1 shows that the equivariant denoiser closely tracks the (supervised) MMSE denoiser, in line with the theoretical prediction. This empirical agreement suggests that, for all three datasets considered, we are operating within the regime described by Theorem 1.
> > >
> > > - Regarding [1], it says "It is worth noting that the proposed DRUNet .... naturally enforces scaling invariance property of many image restoration tasks, i.e., f(ax) = af(x) holds true for any scalar a ≥ 0 (please refer to [64] for more details)." Furthermore, in [64] says "Scaling invariance is intuitively desireable for a denoising method operating on natural images; a rescaled image is still an image.".
> > >
> > > Note that the scale invariance discussed above is closely related to the well-documented spatial scale invariance of natural images and their power-law spectral statistics. As commonly argued in the literature, “the causes of this power-law behavior have been the subject of considerable speculation and debate. One of the most widely held views is that it arises from the scale invariance of the visual world. Scale invariance implies that the statistical properties of images remain unchanged under rescaling. In particular, the power spectrum should preserve its shape under such transformations... Only a power-law spectrum remains invariant in shape under this transformation.” (Simoncelli, E.P. and Olshausen, B.A. (2001). Annu. Rev. Neurosci. 24, 1193-1216). Since the norm |x| is directly related to the integral of the Fourier power spectrum, it inherits corresponding power-law (i.e., polynomial) scaling behaviour.

---

### Official Review · Reviewer_97jy · 2026-03-10

**Soundness:** 3
**Presentation:** 3
**Significance:** 3
**Originality:** 3
**Overall Recommendation:** 5
**Confidence:** 3

**Summary:**

This paper studies the problem of learning from noisy measurements using diffusion models. Inspired by a theoretical observation regarding the scale equivariance of the optimal MMSE denoiser, the authors modify the standard denoising objective by introducing a loss that encourages normalisation equivariance across noise levels. This modification enables the training of denoisers below the measurement noise level using only noisy data. The resulting denoiser can then be used within at the sampling stage to perform image restoration tasks. Under their experimental setup, the authors demonstrate the effectiveness of the proposed approach on several image restoration problems.

**Compliance With Llm Reviewing Policy:**

Affirmed.

**Final Justification:**

The authors have addressed my concerns thus I have decided to raise my score.

**Key Questions For Authors:**

- Could the authors clarify the deferences of their paper to the paper from [1] ? Is the third term of the loss for linear inverse problems (Line383 )  related to the idea of "further corruption" introduced in [1]?-
- Could the authors comment on whether the proposed method would scale to larger datasets such as ImageNet, as well as to higher-resolution images beyond the setting considered in the experiments?
- The paper assumes that all samples are corrupted using the same measurement operator A. Could the authors comment on how realistic this assumption is in practice?

**Limitations:**

yes

**Strengths And Weaknesses:**

**Strengths**
- The paper studies a practically relevant problem: training generative models when no clean data are available. This setting arises in many real world application where acquiring high quality data can be expensive or infeasible.
- The idea of leveraging equivariance to learn a denoiser below the measurement noise level is well motivated. The authors also support this intuition with a theoretical result.


**Weaknesses**

- The problem studied appears closely related to prior work such as [1]. Providing a clearer discussion of the differences between the two approaches would help better position the contribution within the existing literature.
- The experimental setup is somewhat limited. The method is only evaluated on medium sized datasets (FFHQ, AFHQ) at a resolution of 128 \times 128. It is unclear whether the approach would generalise to larger datasets or higher resolutions.
- The experiments rely on synthetically generated noise and degradation models. It is unclear how the proposed method would perform in more realistic settings.


**References**
- [1] Daras, G. et al. Ambient diffusion: Learning clean distributions from corrupted data. Advances in Neural Information Processing Systems, 2023.

---

> ### Author Rebuttal · Authors · 2026-03-31
>
> We appreciate the reviewers constructive feedback on our paper.
>
> (1) Connection to Ambient Diffusion
>
> While both Ambient Diffusion and the proposed method aim at learning from incomplete measurement data, they crucially vary in i) the assumptions on the operator and noise, and ii) the self-supervised loss used to train the reconstruction network.
>
> - Ambient Diffusion relies on access to noiseless measurements associated with varying forward operators, whereas our method handles noisy measurements associated to a single forward operator.
>
> - The Ambient Diffusion loss ``recorrupts'' the input measurements to learn in the nullspace of the operators, whereas the proposed loss learns in the nullspace of a single operator by enforcing equivariance to transformations via the equivariant imaging term.
>
>
> (2) Larger datasets + image sizes
>
> Our original submission includes experiments  on images of size $128 \times 128$ from moderately sized datasets. We have no reason to believe our method cannot scale to larger data sizes at training and inference time. At inference time, there is no difference in cost compared to supervised denoising diffusion methods. At training time, the cost is equivalent to supervised denoising diffusion methods plus the additional cost of one denoiser evaluation per denoising step due to the computation of the divergence operator, which is not prohibitively costly. Moreover, we explored the scaling of training set size within the AFHQ dataset (see Figure 4). We noted that the denoiser performance followed $C/N^{1/3}$, which is close to the complexity typically observed in supervised learning Hardt (2022), suggesting our method should also track supervised denoising model performance on larger datasets.
>
>
> (3) Application to real data
>
> This is an excellent point from the reviewer. These self-supervised methods are not very useful unless validated on real data. To this end, we provided experiments on noisy MRI scans acquired on our own lab MRI scanner in Appendix section A.11 (see Figure 24). The data was acquired on a permanent magnetic scanner, which has significantly lower SNR than higher field scanners. The data used in these experiments is inherently very noisy (see Figure 23), and we did not add any synthetic noise for our experiments. We trained and tested our method directly to the real data after estimating the noise level. Our results show that our method does perform well on this real data. We don't provide any numerical metrics for these experiments because there are no ground truth samples from this dataset.
>
>
> (4) Real world motivation
>
> There are many scenarios where all measurements are acquired using a single operator:
>
> - On most clinical MRI scanners and clinical MRI protocols, the undersampling pattern is predefined Yaman (2020). While the user could potentially modify the MRI pulse sequence, this is not typically done by technicians operating the scanners.
>
> - Sensors in industrial and medical CT scanners are typically placed at fixed positions, which cannot be easily modified to accommodate new configurations or sensing angles.
>
> - In the case of image demosaicing, we expect a fixed sensor with a fixed arrangement of pixel measurements, which is usually determined before the construction of the sensor, as opposed to an adaptive arrangement.

---

> > ### Author Rebuttal · Reviewer_97jy · 2026-04-02
> >
> > Thank you to the authors for their response. I appreciate the detailed answers and the effort to address the concerns raised. Most of my questions have been clarified. However, the method has not yet been evaluated on larger datasets or higher-resolution images, which would further strengthen the paper. Therefore, I choose to maintain my score.

---

> > > ### Author Response · Authors · 2026-04-03
> > >
> > > Following your suggestion, we repeated training of both our proposed method as well as SURE on ImageNet at a larger resolution (256x256). Training speeds for both methods were approximately 3.8x slower than training at 128x128 due to the larger image sizes. Performance of the denoisers tracked our results on the other datasets by showing improved perceptual quality over SURE (See results below). Qualitative results are similar: our method shows improved perceptual quality compared to slight blurring from SURE (we will include these results in the final version if accepted). We note that we did not conduct this experiment using the comparison method Consistent Diffusion (CD), because because CD takes a prohibitive amount of time during training due to the need to run a sampler at training time making it more difficult to scale to higher image resolutions than our method. We note that the CD paper trained models from scratch only on 64x64 image sizes.
> > >
> > > SURE, LPIPS: $0.0566 $   FID: $22.0077$
> > >
> > > OURS, LPIPS: $\mathbf{0.0506}$    FID: $\mathbf{16.8193}$
> > >
> > > we hope this helps answer the reviewers concerns with using our method at larger resolutions and dataset sizes.

---

### Official Review · Reviewer_2n5Z · 2026-03-12

**Soundness:** 3
**Presentation:** 2
**Significance:** 3
**Originality:** 3
**Overall Recommendation:** 4
**Confidence:** 2

**Summary:**

The authors propose a clever framework to overcome limitations of the SURE estimate
for lower noise levels under observations of a single forward operator by taking advantage
of the reverse-time diffusion SDE. Their results are promising and validated in a set of tasks, settings and modalities that are adequate given the methodological focus of this contribution.
Notably, compared to existing methods, their method seems to perform very competitively
to a supervised counterpart (access to original x, and not noisy or rank-deficient measurements).

**Compliance With Llm Reviewing Policy:**

Affirmed.

**Key Questions For Authors:**

- Is there other more recent methods beyond SURE worth noting and comparing the baseline methods to?
- Future work seems mostly empirical, this is ok given the theoretical contribution, but I would prefer if the text spent more time making the introduced concepts a bit more digested. Since the experiments are limited, perhaphs a lot of the experimental section may be best relagated to the Appendix.
- Is there a good reason not to have an Appendix with adequate sections? The deadline for the Appendix was the same as for the final submission.

**Limitations:**

No access to Appendix to validate properly.

**Strengths And Weaknesses:**

Soundness
The Appendix was missing, so a lot of the claims and references in the main text could not be properly validated. Assumptions made in Equation 7 are not properly motivated. Before equation 8 the authors claim that equivariance can be imposed by constraining the architectures, but fail to motivate their proposed alternative.

Understanding normalization Invariance is a welcome subsection, but could benefit from clearer notation.

The link between $\|x\|$ not conveying information about the normalized signal is adequate, but the power-law, although plausible, should be better motivated.


Presentation
The Appendix is not included nor is it adequately referenced in the main text. The Appendix should be
properly formatted with its own sections and subsections, and references in the main text should point to it.

The notation can be a bit dense, and the assumptions are sometimes hard to follow solely by access to the main text.  There are Figures being referenced that do not exist, or maybe were left in the Appendix, like Figures 21, 15, 16.
Theorem 3.1 is quite involved and should be better introduced. Its proof can be relegated to the appendix, leaving space to include additional Figures or Tables in the main text.

---

> ### Author Rebuttal · Authors · 2026-03-31
>
> We thank the reviewer for their time and thoughtful feedback on our paper.
>
> (1) Issues with Appendix
>
> We apologize for the confusion about experiments/references to the Appendix. The appendix was submitted by the required deadline, as a separate PDF document, available to reviewers as Supplementary Material via OpenReview. Please let us know if you cannot access this file or if it does not satisfactorily clarify the issues regarding the references and claims in the main text linking to the contents of the appendix. In the appendix, we included many additional results/experiments along with proof elements that accompany the main paper.
>
> (2) Clarify Assumptions in Eq. 7
>
> The property in Eq. 7 relies on the assumption that any scale and shifted copy of the image $\mathbf{x}$, $\mathbf{x'}=
> \alpha \mathbf{x} + \mu \mathbf{1}$, is also an admissible clean image. Then, the scaled and shifted noisy measurement $\alpha \mathbf{y} +\mu \mathbf{1} = \mathbf{x}' + (\alpha\sigma) \boldsymbol{\eta}$ corresponds to a noisy measurement of this clean image, with noise level $\alpha \sigma$ instead of $\sigma$. We have included an explanation in the updated manuscript.
>
>
> (3) Clarify Theorem 3.1
>
>  Theorem 3.1 states, in essence, that Gaussian MMSE denoisers are approximately equivariant to rescaling in a manner that is particularly useful for self-supervised learning from noisy data. Specifically, these denoisers can be shown to lie close to denoising operators that are exactly scale-invariant, with the degree of this closeness governed primarily by general structural properties of the prior that are widely observed in natural signals. This result is significant because it indicates that accurate surrogates for the MMSE denoiser can be trained, in a fully self-supervised manner, for a broad range of noise levels below measurement noise. Figure 1 provides strong empirical support for this claim. This then enables training modern denoising score-matching models directly from noisy data, as demonstrated across Section 4. We will make this clearer in the final version of our paper.
>
> (4) Loss vs Architecture Design choice
>
>  As the reviewer points out, we had a choice to enforce equivariance via NN architecture or using our loss function. We show experimentally in Appendix section A.10 (see Figures 21 and 22) that, in the considered self-supervised setting, enforcing equivariance via architecture as had been done in previous supervised approaches, produces worse results than our loss-based appraoch. We believe this to be due to the strict constraints this places on NN architecture components. In contrast,  enforcing equivariance via the loss function, does not impose such constraints and allows using SOTA denoising architectures.
>
> (5) Clarity of Normalization Invariance
>
>  We appreciate the feedback from the reviewer on the clarity of our introduction to normalization invariance. We completely agree that this section could be better introduced and we will make this clearer in the final version of our paper.
>
> (6) Motivating Power-law
>
>  There is a well-established literature demonstrating that natural images exhibit power-law behavior in their energy and spectral statistics. See, e.g., Ruderman, D.L. (1994) Network: Computation in Neural System, 5(4), pp.517-548; Ruderman, D.L. and Bialek, W. (1994). Phys. Rev. Lett. 73, 814-817; Simoncelli, E.P. and Olshausen, B.A. (2001). Annu. Rev. Neurosci. 24, 1193-1216. This phenomenon is closely related to the self-similar nature of natural signals. For completeness, Figure 14 in the Appendix illustrates this property, showing that the distribution of image patch norms is nearly invariant to patch size, up to a constant scaling factor.
>
> (7) Comparison Methods
>
>  There are two cases to consider: (1) measurements corrupted by noise only, and (2) measurements corrupted by a degradation operator + noise. In the case of (1), we compared to SURE, but also to a prior SOTA method for unsupervised denoising Daras (2024), which we outperform. There are other frameworks for self-supervised denoising, such as Noiser2noise Moran (2019), but this has been shown to be equivalent to SURE Monroy (2025) and thus covered by our comparison with SURE. In the case of (2), we compared to the SOTA self-supervised generative method AmbientDiffusion Daras (2023) which assumes access to multiple measurement operators at training time (leading to degraded performance in the single operator setting), and a SOTA point estimator EI (built for self-supervised learning in the single operator setting) Chen 2022.
>
> (8) Reallocate space for clarifying concepts
>
>  We hope that our answers to the questions above have helped to clarify different aspects of the paper and we will incorporate these changes into the final version of the paper to enhance clarity. Specifically, moving some results to the appendix to make room for clearer descriptions of the relevant concepts our method builds upon.

---

> > ### Author Rebuttal · Reviewer_2n5Z · 2026-04-03
> >
> > Fully resolved.

---

> > > ### Author Response · Authors · 2026-04-03
> > >
> > > Thank you for acknowledging our rebuttal and letting us know that your comments are fully addressed. We kindly ask that you reconsider your final score to reflect your agreement with the discussion and changes we made.

---

### Decision · Program_Chairs · 2026-04-30

**Decision:**

Accept (regular)

**Comment:**

The reviewers found the approach well-motivated, with strong empirical results and meaningful improvements over existing self-supervised methods. Most concerns were satisfactorily addressed in the rebuttal. Overall, the paper makes a solid contribution.